# Replication dynamics identifies the folding principles of the inactive X chromosome

**Rawin Poonperm** [1], **Saya Ichihara** [2,5], **Hisashi Miura**[1], **Akie Tanigawa**[1], **Koji Nagao**[3], **Chikashi Obuse**[3], **Takashi Sado** [2,4] & **Ichiro Hiratani** [1] ✉

Chromosome-wide late replication is an enigmatic hallmark of the inactive X chromosome (Xi). How it is established and what it represents remains obscure. By single-cell DNA replication sequencing, here we show that the entire Xi is reorganized to replicate rapidly and uniformly in late S-phase during X-chromosome inactivation (XCI), reflecting its relatively uniform structure revealed by 4C-seq. Despite this uniformity, only a subset of the Xi became earlier replicating in SmcHD1-mutant cells. In the mutant, these domains protruded out of the Xi core, contacted each other and became transcriptionally reactivated. 4C-seq suggested that they constituted the outermost layer of the Xi even before XCI and were rich in escape genes. We propose that this default positioning forms the basis for their inherent heterochromatin instability in cells lacking the Xi-binding protein SmcHD1 or exhibiting XCI escape. These observations underscore the importance of 3D genome organization for heterochromatin stability and gene regulation.

In mammals, one X chromosome is inactivated in females to equalize X-linked gene expression relative to males during early development[1,2]. Microscopic studies have revealed that this Xi is more compact than the active X (Xa)[3,4], leading researchers to explore the relationship between three-dimensional (3D) genome organization and transcription[5].

A genome-wide chromosome conformation capture technology, Hi-C[6], has revealed that mammalian autosomes are composed of Mb-sized topologically associating domains (TADs)[7,8]. TADs can be in either active (A) or inactive (B) nuclear compartments[6], which are further subdivided into several subcompartments[9,10]. It was reported that when the Xi forms during mouse embryonic stem cell (mESC) differentiation, neighboring small A or B compartment domains on the Xi initially fuse with each other to form larger S1 or S2 compartment domains, which further merge to form the compact Xi structure through the actions of SmcHD1 (ref. 11), a global Xi-binding protein known for its role in X-chromosome inactivation (XCI) maintenance[11–15]. Unlike the autosomes, Xi lacks TADs and compartments but instead has a unique bipartite 'megadomain' structure separated by a tandem

macrosatellite repeat, *Dxz4*/*DXZ4* (refs. 9,16–18). While many XCI regulators have been identified, how the Xi acquires this unique, compact 3D organization is still unclear[2,5].

An evolutionarily conserved chromosome-wide early-to-late (EtoL) replication timing (RT) switch of the Xi has long been known[19–22]. While its biological significance remains obscure, a tight relationship between RT, A or B compartments[23–25] and subcompartments[10] suggests that Xi's RT dynamics might reflect its compartmentalization. However, Xi's RT regulation has been studied primarily by microscopy or bulk genome-wide replication assays[13,19,25–28]. Here we used bulk and single-cell DNA replication sequencing[29–31] to address the Xi's dynamic RT changes during mESC differentiation. Haplotype-resolved 4C-seq revealed that the Xi's RT indeed reflected its compartment organization. Because SmcHD1 is involved in the Xi's RT and compartment regulation in humans[12] and mice[11,13–15], we also used SmcHD1-mutant cells. Our results are consistent with the idea that the default 3D architecture of the X chromosome forms the basis for regional differences in Xi heterochromatin stability.

[1]Laboratory for Developmental Epigenetics, RIKEN Center for Biosystems Dynamics Research (BDR), Kobe, Japan. [2]Department of Advanced Bioscience, Graduate School of Agriculture, Kindai University, Nara, Japan. [3]Department of Biological Sciences, Graduate School of Science, Osaka University, Toyonaka, Japan. [4]Agricultural Technology and Innovation Research Institute, Kindai University, Nara, Japan. [5]Present address: Cell Architecture Laboratory, Department of Chromosome Science, National Institute of Genetics, Shizuoka, Japan. ✉e-mail: ichiro.hiratani@riken.jp

## Results

### An mESC-based system to study the RT dynamics of the Xi

To explore the RT dynamics of the Xi as it forms, we need to distinguish the two Xs by single nucleotide polymorphisms (SNPs). We used an F1-hybrid mESC line, JB4/EI7HZ2, from a cross between JF1 female and B6 male[32] (Fig. 1a). In JB4/EI7HZ2 mESCs, *CAGzeo* is on the JF1-X at the *Hprt* locus and *IRESneo* is inserted on the B6-X in the 3' untranslated region of *Eif2s3x*, an XCI escapee. In G418[+]/zeocin[+] medium, XO cells lacking either of the drug-resistance genes are eliminated and the XX cells show 100% skewed XCI with B6-X being the Xi, because differentiated cells with JF1-Xi cannot survive due to *CAGzeo* inactivation (Fig. 1a and Supplementary Fig. 1a). Cells with B6-Xi can survive because the escapee locus allows *IRESneo* expression from the Xi. We used two neural differentiation protocols to compare mESCs ('before XCI'), day 7 or 9 early neurectoderm cells ('during XCI')[25], and neural stem cells (NSCs) obtained after roughly 3 weeks of differentiation ('after XCI')[33] (Fig. 1a). Their differentiation states were confirmed by PCR with reverse transcription, immunofluorescence and RNA-sequencing (RNA-seq) (Supplementary Fig. 1a–e). Day 7 or 9 neurectoderm cells and NSCs showed roughly 75 and 90% *Xist* RNA cloud formation, respectively, suggesting nearly uniform differentiation and XCI (Supplementary Fig. 1f).

### The Xi becomes late replicating during mESC differentiation

Our routine genome-wide RT assay (Repli-seq) is based on BrdU immunoprecipitation (BrdU-IP) from early and late S-phase cells fractionated by flow cytometry, followed by next-generation sequencing (NGS) (Fig. 1b). Then, relative enrichment of early- and late-replicating DNA is analyzed to generate a genome-wide RT map. Repli-seq data of mESCs, day 7 or 9 neurectoderm and NSCs were highly reproducible (Supplementary Fig. 2a). Because Repli-seq allows cell-type profiling[25,27,34], we compared our Repli-seq data with various other datasets[30] and confirmed distinct and proper differentiation states (Fig. 1c).

Haplotype-resolved Repli-seq of mESCs, days 7 or 9 neurectoderm and NSCs revealed that the B6-X is converted to a chromosome-wide late-replicating Xi by day 7 and maintained thereafter, while the JF1-X remains active and maintains similar RT profiles (Fig. 1d and Supplementary Fig. 2b). We classified the ESC-to-NSC RT regulation into four groups based on the mean RT values of each 400-kb bin (Fig. 1d and Supplementary Fig. 2c): early-to-early (EtoE, mean RT of more than zero in both), EtoL (mean RT of more than zero and less than zero in mESCs and NSCs, respectively, with a greater than 0.5 difference), late-to-early (LtoE, mean RT of less than and more than zero in mESCs and NSCs, respectively, with more than 0.5 difference) and late-to-late (LtoL, mean RT of less than zero in both). In contrast to the JF1-X, almost all early RT bins on the B6-X in mESCs became late in NSCs (99 out of 102 EtoL bins of 39.6 Mb), while EtoE and LtoE domains were nearly nonexistent (Fig. 1d and Supplementary Fig. 2c). Analysis of CBMS1 mESCs using the same differentiation protocol found an EtoL RT change of the Xi during days 5–7 when the cells acquired a late epiblast fate[25], consistent with the emergence of a late-replicating Xi in

the postimplantation epiblast[19]. The Xi's EtoL RT change probably also occurs around days 5–7 of JB4/EI7HZ2 mESC differentiation, although we did not pursue this further (as distinguishing gradual RT changes in all cells versus cells with abrupt RT changes that were gradually increasing was a challenge).

### RT delays and advances generate a uniformly late RT Xi

To further dissect the Xi's replication kinetics, we performed single-cell Repli-seq (scRepli-seq) with cells throughout the S-phase to construct what we call the 'whole-S' RT profiles (Fig. 1e). In NSCs, the whole-S RT profile of the JF1-Xa looked similar to the autosomes, as expected (Fig. 1e and Supplementary Fig. 2d). By contrast, the B6-Xi initiated replication in the second half of S-phase and lacked clearly distinguishable early or late RT patterns (Fig. 1e).

Unlike the Xa, the replication score (percentage of replication) progression of the Xi was uncoordinated with the autosomes (Fig. 1f,g and Supplementary Fig. 2e,f). The steep rise in the Xi's replication score of the fitted sigmoid curve (Fig. 1f, red line based on scRepli-seq average) suggested fast completion of the Xi replication with a $T_{10-90\%}$ (defined as the time required for a chromosome to go from 10 to 90% replication, assuming a 10 h S-phase) of 3.4 h. By contrast, the $T_{10-90\%}$ of the Xa and the autosomes were significantly longer, on the order of 8 h (Fig. 1f and Supplementary Fig. 2e). The replication scores of the Xi were variable among cells after 60–70% S-phase (Fig. 1g). This suggests large variability in the timing of replication initiation among cells and, in turn, the timing of replication completion of the Xi. Thus, the Xi's $T_{10-90\%}$ of 3.4 h is probably an overestimate.

While widespread EtoL RT changes were expected, the Xi's latest-replicating regions advancing their RT toward mid-late S in NSCs was unexpected (Fig. 1h), as they were classified as LtoL domains based on Repli-seq (Fig. 1d and Supplementary Fig. 2c). Thus, scRepli-seq revealed that the Xi's RT change is chromosome wide, with both advances and delays to achieve its uniform RT in the second half of S. Although the Xi's RT is uniform and synchronous for a given cell, there is variability among cells with regards to when the Xi initiates replication, which is out of phase with the rest of the genome as if the Xi is disengaged from the genomic RT program.

### SmcHD1 is required for maintaining the uniformly late RT Xi

SmcHD1 is required for Xi's late RT in human TERT-RPE1 (hTERT-RPE1) cells, mouse embryonic fibroblasts (MEFs) and mouse embryos[12–14]. To see whether SmcHD1 also affects Xi's RT during mESC differentiation, we generated SmcHD1-mutant mESCs by CRISPR–Cas9 with a guide RNA targeting its ATPase domain. We confirmed 5- and 10-bp deletions, premature stop codons biallelically on exon 3 and SmcHD1 loss by western blotting (Supplementary Fig. 1g,h). Differentiation states of SmcHD1-mutant and wild-type (WT) cells were similar (Supplementary Fig. 1a–f), and so were the RT profiles of their Xs in mESCs and day 7 or 9 cells (Fig. 2a and Supplementary Fig. 3a–d). Thus, SmcHD1 was largely dispensable for the initiation of Xi's EtoL RT change, as in mouse

**Fig. 1 | The Xi replicates rapidly and uniformly in late S during XCI. a**, Neural differentiation protocols of JB4/EI7HZ2 mESCs. **b**, Schematic diagram of BrdU-IP Repli-seq experiment. BrdU-labeled cells are sorted into early and late S-phase fractions (gating strategy is shown in Supplementary Fig. 1i), which are subject to immunoprecipitation (IP) using an anti-BrdU antibody. The ratio of early and late S-phase DNA is calculated to generate a genome-wide RT profile. **c**, Comparison of genome-wide RT profiles (sliding windows of 200 kb at 80-kb intervals, excluding the X) during differentiation of JB4/EI7HZ2 mESCs, CBMS1 mESCs[30], EpiSCs and MEFs by hierarchical clustering. **d**, BrdU-IP RT profiles of the JF1-X and the B6-X during JB4/EI7HZ2 mESC differentiation (average of 2–3 replicates, 400-kb bins). RT classes represent four types of ESC-to-NSC RT regulation, which were described on the right. The blue line shows *Dxz4* position. **e**, Binarized whole-S scRepli-seq profiles of the JF1-Xa and the B6-Xi from 94 NSCs throughout the S. The scRepli-seq profiles are ordered by their percentage replication scores

(of the whole genome excluding the X). BrdU-IP RT of the NSCs is shown for comparison. The blue lines show *Dxz4* position. **f**, Percentage replication scores of the JF1-Xa and the B6-Xi were plotted against those of the whole genome for each NSC (open circles). Cells were divided into 18 groups by their percentage replication scores and the average scores of the JF1-Xa and the B6-Xi in each group are shown by filled circles. To minimize errors during data fitting, we added two pseudo-percentage replication scores, 0 and 100%, at the earliest end and the latest end of the S-phase, respectively. We fitted a linear and a sigmoid model to the JF1-Xa and the B6-Xi, respectively, to calculate the $T_{10-90\%}$ values (the time required to go from 10 to 90% replication, assuming a 10 h S). **g**, Cells were divided into ten groups by percentage replication scores and the boxplots show distributions of percentage replication scores of the JF1-Xa and the B6-Xi in each group. **h**, An exemplary latest RT region on the Xa, which advanced its RT on the Xi in NSCs. CBMS1 mESC data are shown for comparison[30].

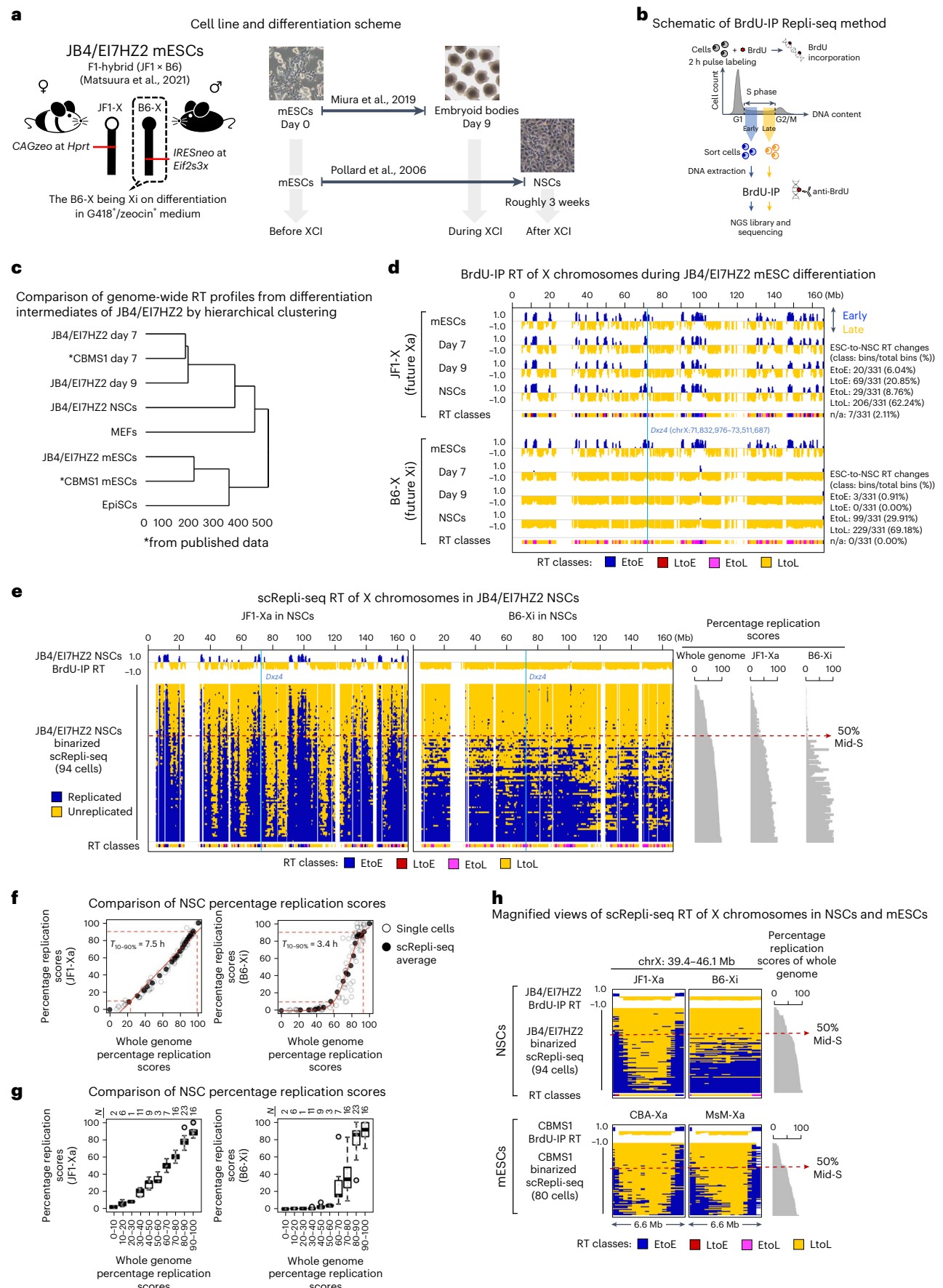

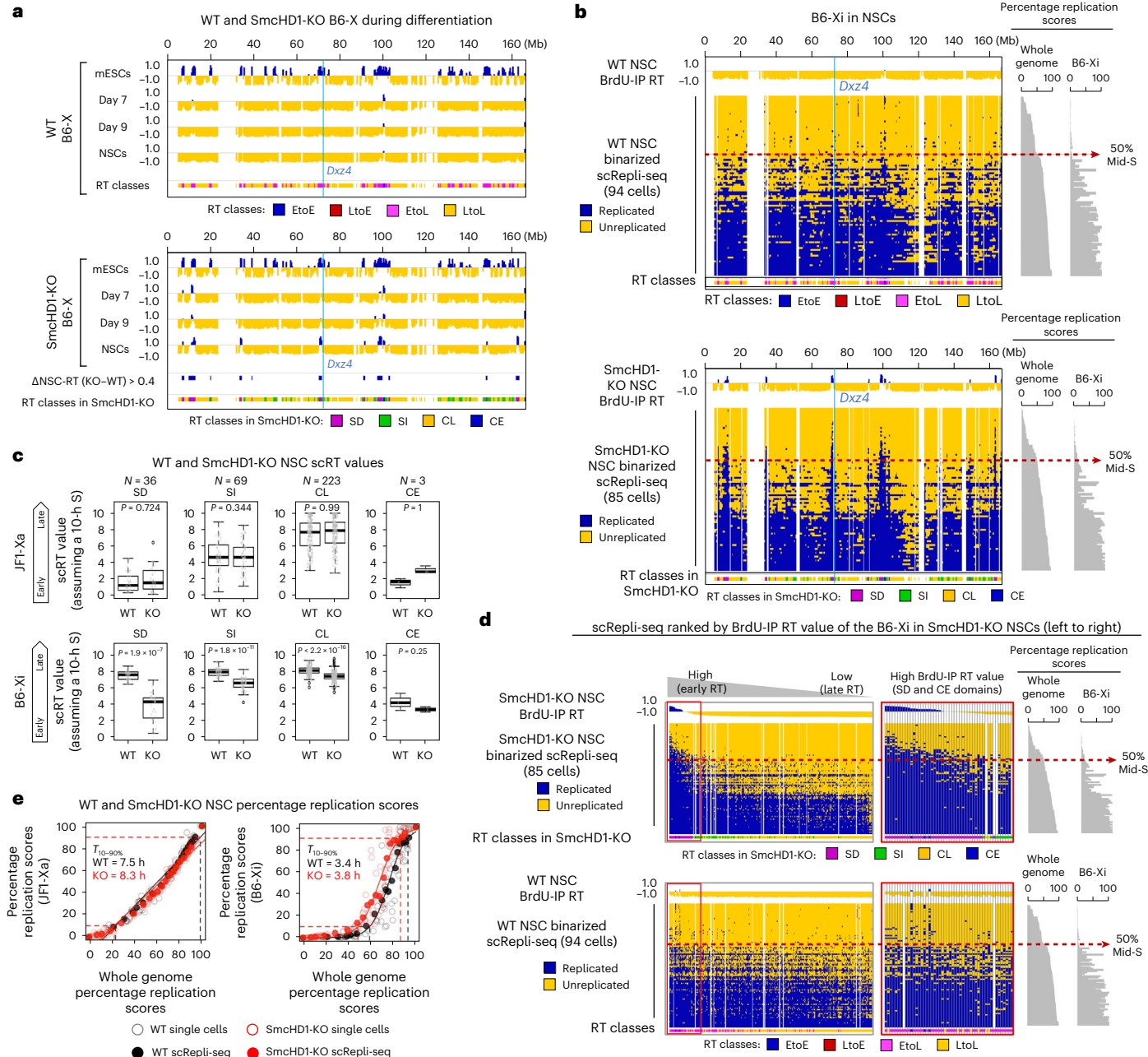

**Fig. 2 | SmcHD1 is required for maintaining the uniformly late-replicating Xi but dispensable for the initiation of the RT switch. a**, BrdU-IP RT profiles of the B6-X during WT and SmcHD1-mutant (KO) mESC differentiation (average of 2–3 replicates). In addition to the ESC-to-NSC RT classes, those identified in SmcHD1-mutant cells are shown. The blue lines show *Dxz4* position. **b**, Whole-S scRepli-seq plots of the B6-Xi from 94 WT NSCs and 85 SmcHD1-mutant NSCs, as in Fig. 1e. ESC-to-NSC RT classes as well as those identified in SmcHD1-mutant NSCs are shown. The blue lines show *Dxz4* position. **c**, Single-cell RT (scRT) values were calculated for four RT classes identified by SmcHD1-mutant NSC analysis.

Single-cell RT represents an estimated time when a given genomic bin replicates in S (assuming a 10 h S). *P* values were obtained from one-sided paired Wilcoxon signed-rank test. *N*, the number of bins in each RT class. **d**, Whole-S scRepli-seq plots in Fig. 2b were sorted by the BrdU-IP RT values of SmcHD1-mutant NSCs (from early to late RT). The red box corresponds mostly to the SD and CE classes and is magnified on the right. **e**, Comparison of percentage replication scores of the Xs in WT (black) and SmcHD1-mutant NSCs (red) with those of the whole genome, as in Fig. 1f.

embryos[14], consistent with SmcHD1's role in XCI maintenance[14,35–37]. By contrast, the B6-Xi had several domains that were earlier replicating in mutant NSCs (Fig. 2a), while the JF1-Xa and autosomes were unaffected (Supplementary Fig. 3a–d). We defined SmcHD1-affected domains as those with an average RT difference of more than 0.4 and found that roughly 10% (36 out of 331 bins, or 14.4 Mb) of the B6-Xi in mutant NSCs exhibited RT reversal (Fig. 2a), most of which were early replicating in mESCs (30 out of 36 bins). Thus, among the EtoL-switching

domains during differentiation to NSCs (99 bins of 39.6 Mb), a subset depends on SmcHD1 for RT maintenance (SmcHD1-dependent EtoL RT domains or SD domains; 30 bins of 12 Mb), while the rest does not (SmcHD1-independent EtoL RT domains or SI domains; 69 bins of 27.6 Mb). The rest of the Xi maintains late replication in mutant NSCs (Fig. 2a, 223 bins of 89.2 Mb of constitutively late (CL) domains and three bins of 1.2 Mb of constitutively early (CE) domains) (Supplementary Table 1).

Using whole-S scRepli-seq, cell-to-cell RT heterogeneity of the SD domains was assessed. The SD domains replicated earlier in SmcHD1-mutant cells (Fig. 2b,c). By sorting the scRepli-seq RT bins according to the average BrdU-IP RT values, we found that most of the mutant NSCs completed the B6-Xi SD-domain replication before mid-S, while WT NSCs initiated SD-domain replication after mid-S (Fig. 2d). Thus, most cells exhibited SD-domain RT reversal without SmcHD1.

Because the SD domains occupied only about 10% of the Xi, the overall Xi replication duration judged by $T_{10-90\%}$ was similar between the mutant and WT NSCs (Fig. 2e). The mutant Xi initiated replication slightly earlier than the WT Xi (Fig. 2c,e). However, the non-SD portion of the mutant Xi still replicated rapidly and uniformly as in WT cells (Fig. 2b,d). Unlike the Xi, the JF1-Xa and the autosomes exhibited similar whole-S scRepli-seq profiles (Fig. 1e and Supplementary Fig. 2d) and coordinated replication progression (Fig. 1f,g and Supplementary Fig. 2e,f), which were maintained in SmcHD1-mutant NSCs as assayed by BrdU-IP RT (Supplementary Fig. 3a–d), whole-S scRepli-seq (Supplementary Fig. 3e,f) and single-cell RT (scRT) values (Supplementary Fig. 3g).

### SD domains protrude out of the Xi core in SmcHD1-mutant NSCs

While RT closely reflects the A or B compartments[23,25] and subcompartments[10], Xi's compartments are elusive[38]. To test whether RT reflected the Xi's compartment organization, we used haplotype-resolved 4C-seq (refs. 39,40) (Supplementary Fig. 4a,b) and analyzed nine viewpoints (Fig. 3a and Supplementary Fig. 4c). The 4C-seq profiles of both Xs in WT and SmcHD1-mutant mESCs looked identical (Fig. 3b and Supplementary Figs. 4d, 5 and 6). Early-replicating (SD, SI and CE) viewpoints interacted with other early-replicating domains, skipping the late-replicating domains in between, while late-replicating CL viewpoints showed interactions primarily within their resident late-replicating domains (Fig. 3b and Supplementary Figs. 5 and 6), consistent with spatial segregation of early- and late-replicating compartments. On day 9, the B6-Xi in WT and SmcHD1-mutant cells still looked similar but showed less contrast between the peaks and valleys, resulting in less significant far-*cis* interactions than in mESCs (Fig. 3b and Supplementary Figs. 4d and 6; far-*cis* contact numbers in the top right corner of each plot). In WT NSCs, the contrast was even weaker, consistent with Xi's reorganized RT (Fig. 3b and Supplementary Fig. 6). In SmcHD1-mutant NSCs, the 4C-seq profiles of the SD viewpoints on the B6-Xi were markedly different and had peaks at SD-domain positions, consistent with their RT reversal (Fig. 3b and Supplementary Figs. 4d and 6). In addition, the SD viewpoints in mutant NSCs showed weaker *cis* interactions with their surrounding regions compared to SI, CL and CE viewpoints (Supplementary Fig. 7). These observations are consistent with the idea that SD domains protrude out of the Xi core and contact each other in SmcHD1-mutant NSCs.

We find much fewer defects for non-SD viewpoints on the mutant NSC B6-Xi (Fig. 3b and Supplementary Figs. 4d and 6; except for the 45-Mb viewpoint (VP45Mb); discussed later). The JF1-Xa in WT and mutant cells maintained similar 4C-seq profiles during differentiation (Supplementary Figs. 4d and 5). These results indicate that the B6-X is gradually transformed into a uniformly compacted structure that lacks segregation into early- and late-replicating compartments during differentiation, confirming earlier studies[11,41]. In SmcHD1-mutant NSCs, however, the SD domains protrude out and contact each other.

### SD domains are prone to reactivation in SmcHD1-mutant cells

SmcHD1-mutant cells exhibit Xi reactivation[11,13,14,35,42]. To examine its relationship to Xi compartmentalization, we reanalyzed RNA-seq data of WT and SmcHD1-mutant neural progenitor cells (NPCs)[11], as their cell identity was close to NSCs (Supplementary Fig. 8a). We compared the gene expression of different RT classes because RT accurately reflected the Xi compartmentalization (Fig. 3b and Supplementary Fig. 6). The SD-domain genes showed the highest reactivation in SmcHD1-mutant

cells (Fig. 4a,b), which largely overlapped with the SmcHD1-sensitive class I genes[11] (Fig. 4a and Supplementary Fig. 8b). The SI domain showed significant reactivation but to a much lesser extent, while the CL and CE domains did not (Fig. 4a,b). Thus, Xi reactivation is strongly correlated with compartmentalization defects.

We further analyzed the relationship of Xi reactivation with epigenetic status. The SD domains coincided with Xi regions in SmcHD1-mutant NPCs showing a significant decrease in *Xist* RNA binding (Fig. 4a,c), a decrease in histone H3 lysine 27 trimethylation (H3K27me3) (Fig. 4a,d) and an increase in H3K4me3 (Fig. 4a,e). The SI genes did not show less *Xist* binding but showed a significant increase in H3K4me3, although to a lesser extent than the SD genes (Fig. 4a,c,e). The SD domains also coincided with regions depleted of H3K27me3 in SmcHD1-mutant MEFs[13] (Supplementary Fig. 8c,d), although parts of the SD domains retained H3K27me3. Several SI domains close to the telomere also showed H3K27me3 depletion (Supplementary Fig. 8d).

We analyzed SmcHD1 enrichment on the Xi using published data[11,13]. SmcHD1 was particularly enriched on the SD domains in MEFs but not NPCs (Supplementary Fig. 8e), suggesting that SmcHD1 binding is cell-type specific. Nonetheless, SD-domain defects in SmcHD1-mutant MEFs and NPCs were similar based on H3K27me3 data (Fig. 4a,d and Supplementary Fig. 8c,d). Thus, the SD domain's susceptibility to SmcHD1 mutation is not due to the preferential binding of SmcHD1 to these domains.

### DNA-FISH showed SD-domain protrusion from the mutant Xi core

To validate the *Xist* RNA binding data, we performed sequential *Xist* RNA fluorescence in situ hybridization (RNA-FISH) and DNA-FISH using two probe sets targeting neighboring SD, SI and/or CL domains (Fig. 5a). We categorized the DNA-FISH signal localization relative to the *Xist* cloud into four groups (Fig. 5b). In WT NSCs, most of the SD, SI and CL signals were positioned similarly close to the Xi surface (Fig. 5b,c,e and Supplementary Fig. 8f) or from the *Xist* cloud centroid (Fig. 5d,f), confirming earlier studies[43,44]. In SmcHD1-mutant NSCs, however, the SD probes frequently protruded out of the *Xist* cloud (Fig. 5c,e and Supplementary Fig. 8f) and became distant (Fig. 5d,f). The SI (but not CL) probes exhibited the same trend but to a lesser extent (Fig. 5c–f).

To validate the protrusion and interaction of SD domains, we performed pair-wise DNA-FISH using three SD probes (Fig. 5g). Simultaneous protrusion of two probes was much more frequent in SmcHD1 mutant than WT NSCs (Fig. 5h,i). Focusing on the 'two-SD protrusion' cells, the 71-SD to 98-SD distance was significantly closer in SmcHD1-mutant NSCs (Fig. 5j,k), consistent with 4C-seq (Fig. 5g, pair-1). The 98-SD to 148-SD distance was similar in SmcHD1-mutant and WT NSCs (Fig. 5j,k), again consistent with 4C-seq (Fig. 5g, pair-2 interacts in both WT and KO, although slightly higher in KO). The 71-SD to 148-SD distance did not decrease in SmcHD1-mutant NSCs (Fig. 5j,k), consistent with their weak interaction by 4C-seq (Fig. 5g, pair-3). Thus, while there is some cell-to-cell heterogeneity, FISH results were consistent with 4C-seq (see Supplementary Text 1 and 2 for further discussion).

### SD but not SI domains on both Xs contact other chromosomes

Is the SD domain positioning on the Xi surface a cause or a consequence of SD gene reactivation in SmcHD1-mutant cells? The answer was brought about serendipitously when we analyzed interchromosomal interactions by 4C-seq (refs. 39,45). In SmcHD1-mutant NSCs, the B6-Xi SD viewpoints frequently contacted other chromosomes, while the non-SD viewpoints rarely did so (Fig. 6a and Supplementary Fig. 9a), consistent with the idea that the SD but not SI and CL domains are on the outermost surface of the Xi, allowing contact with other chromosomes. This was true for the B6-Xi in WT NSCs and, also for the JF1-Xa (Fig. 6a and Supplementary Fig. 9a), suggesting that this SD domain positioning is independent of transcription or the SmcHD1 genotype.

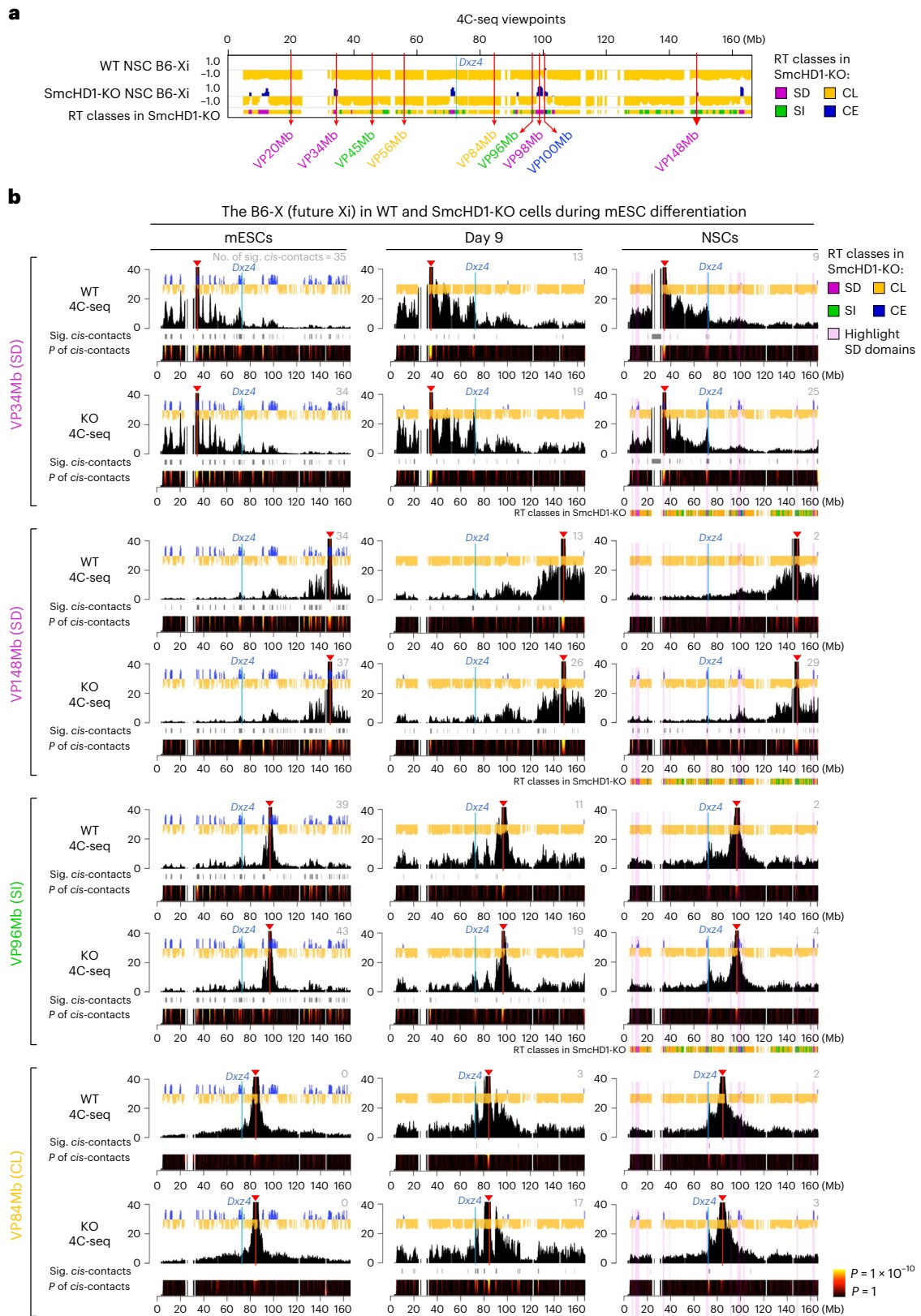

**Fig. 3 | The SD domains protrude out of the Xi core and contact each other in SmcHD1-mutant NSCs. a**, Nine 4C-seq viewpoints (red arrows) on the X. BrdU-IP RT data are shown. Colors represent four distinct RT classes. The blue line shows *Dxz4* position. **b**, Smoothed 4C-seq profiles (black) of the B6-X in WT and SmcHD1-mutant (KO) mESCs (Xa), day 9 differentiated cells (Xi) and NSCs (Xi) overlaid on the BrdU-IP RT profiles (blue, early and yellow, late). Reads from two replicates were combined and plotted in sliding windows of 201 restriction fragments (Methods). Red lines and arrowheads, viewpoints; blue lines, *Dxz4*; gray bars beneath each plot, significant (sig.) far-*cis* contacts (the number of such contacts is shown in the top right corner of each 4C plot in gray); domainogram beneath each plot, the significance of the interaction shown by the color range (window sizes are 2–200 from the bottom to the top); pink highlighted regions, SD domains; colored bars, RT classes.

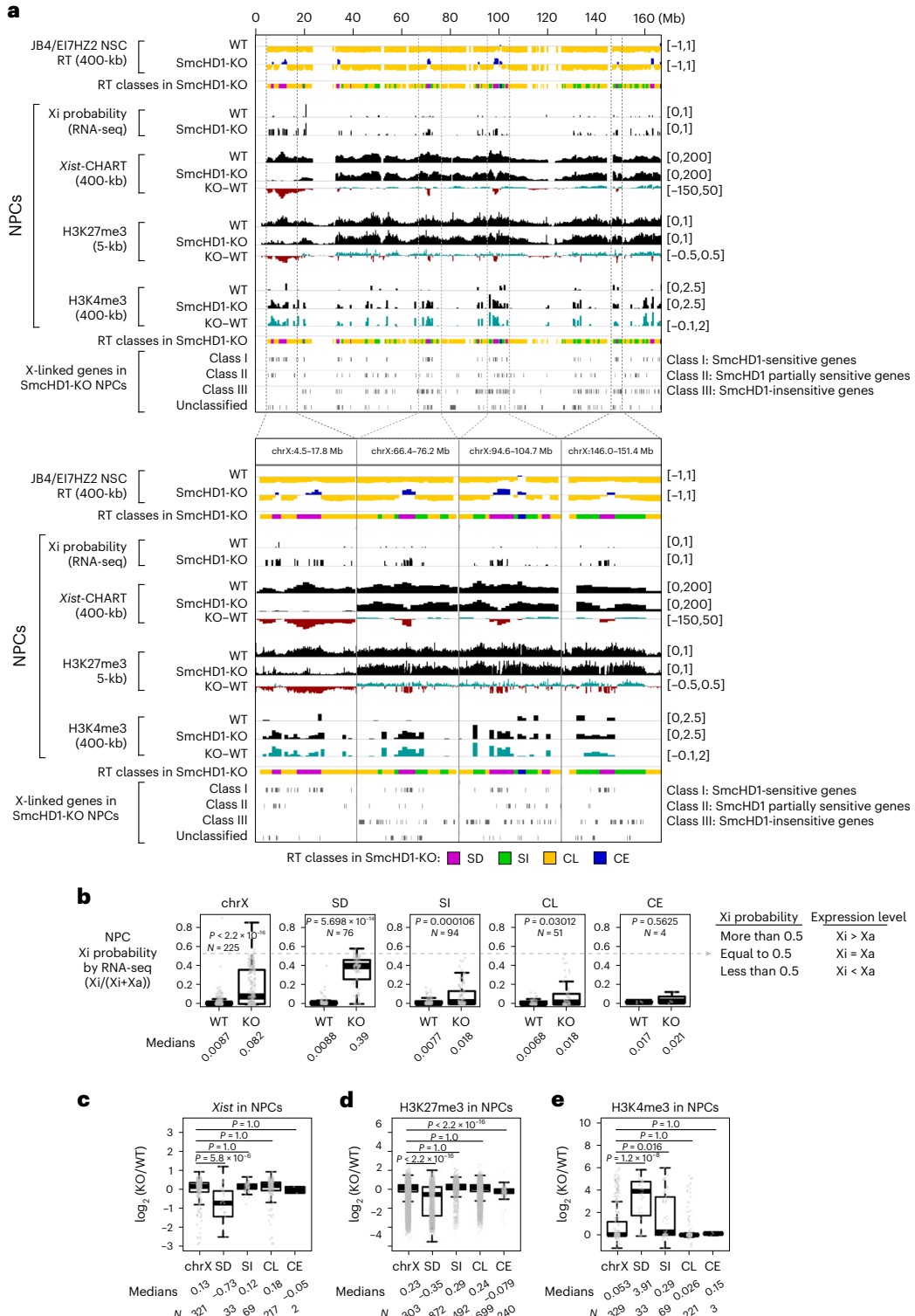

**Fig. 4 | The SD-domain genes are preferentially reactivated and lose repressive Xi signatures in SmcHD1-mutant NPCs. a**, Comparison of Xi probability of X-linked gene expression ((mus-Xi reads)/(mus-Xi reads + cas-Xa reads); 0 and 0.5 represent fully silenced and reactivated Xi states, respectively), *Xist* RNA enrichment by CHART-seq and H3K27me3 and H3K4me3-ChIP–seq (ref. [11]) on the Xi in WT and SmcHD1-mutant (KO) NPCs. BrdU-IP RT of the NSC Xi (B6-Xi) is shown for comparison. Class I (SmcHD1-sensitive), class II (partially SmcHD1-sensitive), class III (SmcHD1-insensitive) and unclassified genes are based on expression analysis by Wang et al.[11]. Bin sizes are shown in parentheses. Upper and lower panels show the entire Xi and the magnified views of four representative SD-domain regions, respectively. Colored bars, RT classes. **b**, Comparison of Xi probability of 225 X-linked gene expression between RT classes in WT and KO NPCs. Escapees in NPCs identified by Wang et al.[11] were removed before the analysis. *N*, the number of X-linked genes in each class. *P* values were obtained from one-sided paired Wilcoxon signed-rank test. **c–e**, Comparison of *Xist* (**c**), H3K27me3 (**d**) and H3K4me3 (**e**) enrichment on the SmcHD1-KO versus WT Xi among RT classes. *N*, the number of genomic bins in each class. The log₂(fold change) is shown. *P* values in **c–e** were obtained from a one-sided Wilcoxon signed-rank test with the Bonferroni correction.

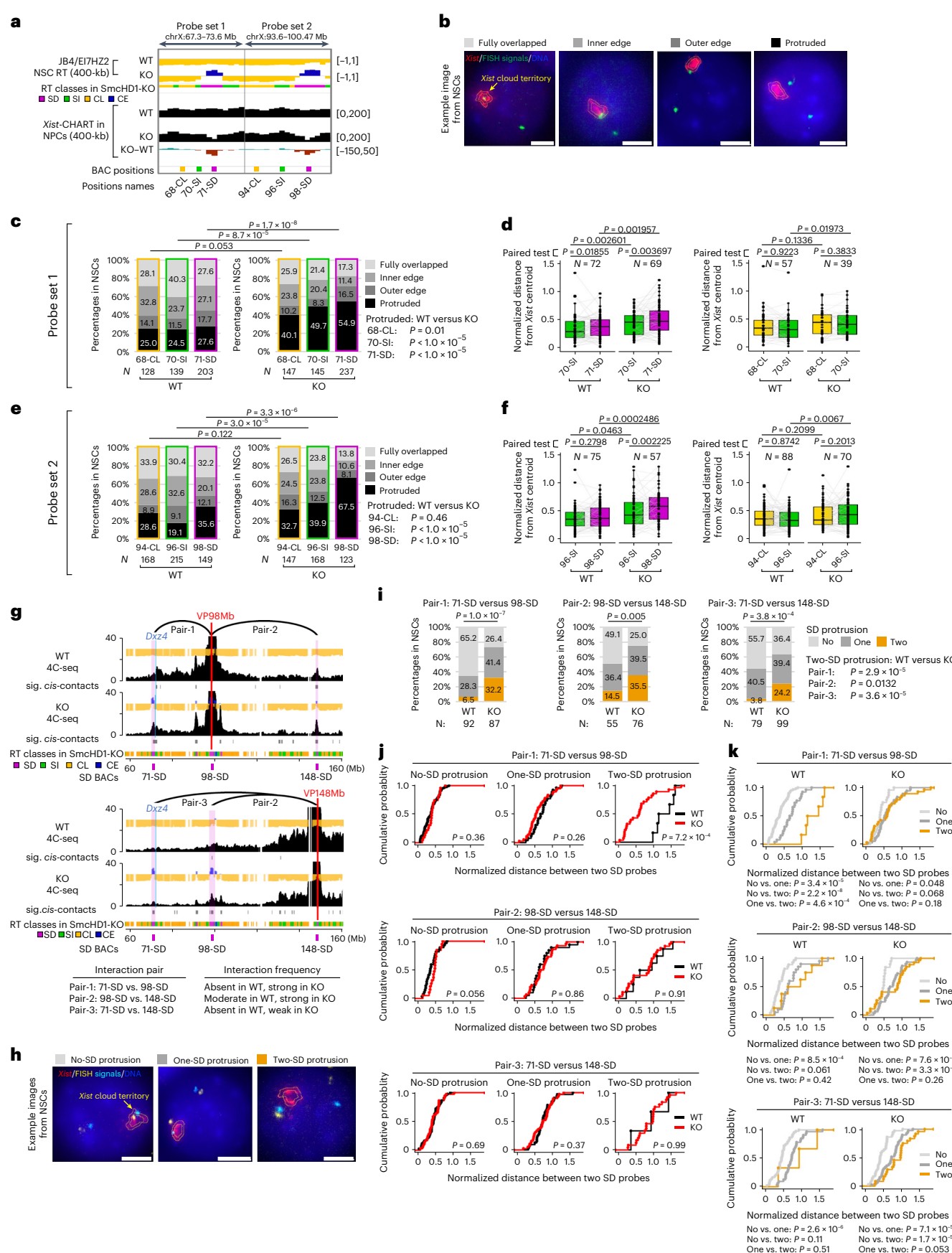

**Fig. 5 | Validation of the protrusion of SD domains out of the *Xist* territory and their closer proximity in SmcHD1-mutant NSCs by FISH. a**, SD, SI and CL domain BACs were used to examine their subnuclear localization relative to the *Xist* cloud. **b**, Representative images of RNA- and DNA-FISH using *Xist* RNA and SD, SI or CL BAC probes in JB4/EI7HZ2 NSCs. We categorized the DNA-FISH signal localization relative to the *Xist* cloud into four groups (Methods). Scale bars, 5 μm. **c,e**, Percentages of DNA-FISH signal localization relative to the *Xist* cloud in NSCs: set 1 (**c**) and set 2 (**e**). **d,f**, Distance between the DNA-FISH signal and the *Xist* cloud centroid in NSCs normalized with *Xist* Feret diameter in the same nucleus: set 1 (**d**) and set 2 (**f**). **g**, Three SD BACs and their interaction frequencies as observed by 4C-seq (Fig. 3b and Supplementary Fig. 6). Red lines, viewpoints; blue lines, *Dxz4*; gray bars beneath each plot, significant far-*cis* contacts; colored

bars, RT classes. **h**, Representative images of RNA- and DNA-FISH using *Xist* RNA and SD BAC probes in JB4/EI7HZ2 NSCs. The SD–SD localization patterns were categorized into three groups. Scale bars, 5 μm. **i**, Percentages of SD–SD localization patterns in WT and SmcHD1-mutant NSCs. **j,k**, Normalized distances between two SD probes were plotted as cumulative plots: comparing WT versus SmcHD1-mutant NSCs (**j**) and comparing the SD protrusion group (**k**). *N*, the total number of cells analyzed from two to three independent experiments. *P* values in **c**, **e** and **i** were obtained from a chi-square test for all groups and a Fisher's exact test for the protruded group. *P* values in **d** and **f** were obtained from one-sided Wilcoxon signed-rank test, with data that used a paired test indicated. *P* values in **j** and **k** were obtained from a two-sided Kolmogorov–Smirnov test.

FISH experiments suggested that the SD, SI and CL domains are equally close to the Xi surface (Fig. 5c–f), possibly due to the resolution limit. Based on the interchromosomal interaction frequency, we predict that the SI and CL domains are just underneath the SD-domain layer in NSCs, avoiding contact with other chromosomes. The B6-Xi SD domains in mutant NSCs make more interchromosomal contacts than in WT NSCs (Fig. 6a and Supplementary Fig. 9a), possibly reflecting their protrusion out of the Xi (Fig. 5c–f,i).

A similar interchromosomal interaction trend was observed for the mESC (Fig. 6b and Supplementary Fig. 9a). To test whether this is true chromosome wide, we performed virtual 4C to analyze interchromosomal interaction using Hi-C data. We analyzed 'virtual' viewpoints throughout the Xs using mESC and NPC Hi-C data[11], after confirming the remarkable similarity of actual and virtual 4C profiles (Supplementary Fig. 9b).

Virtual 4C supported the layered X-chromosome organization in NPCs, with the SD and CE domains showing the strongest interchromosomal interaction, followed by SI, then CL domains on both Xs (Fig. 6c,d; Xi's CE contains *Xist*, which is highly expressed; Supplementary Text 3). The SD domains in SmcHD1-mutant NPCs showed higher interchromosomal interaction frequency than in WT NPCs (Fig. 6c,e), consistent with their protrusion out of the Xi core in the mutant. Such an interchromosomal interaction trend was conserved in mESCs (Fig. 6c,d).

However, we found that the SI viewpoints, especially VP45Mb, and the CE viewpoint showed more interchromosomal interactions in mESCs than NSCs on both Xs (Fig. 6a–c and Supplementary Fig. 9a). As mentioned earlier, VP45Mb was an exceptional SI viewpoint that resembled the SD viewpoints' intrachromosomal interaction pattern, acquiring interactions with other SD domains in SmcHD1-mutant NSCs (Supplementary Fig. 6). Therefore, VP45Mb could be a marginal SI viewpoint located close to the SD layer, showing frequent long-range intra- but not interchromosomal interactions in SmcHD1-mutant NSCs and NPCs. Likewise, the SD and CL layers could be further subdivided.

The viewpoint near VP100Mb maintained frequent interchromosomal interactions in mESCs and NPCs by virtual 4C (Fig. 6c) but not actual 4C (Fig. 6a,b), possibly due to the difference in the viewpoint size and resolution (virtual 4C, 0.4 Mb; actual 4C, 8.6 kb). In addition, some SD or SI virtual viewpoints showed interchromosomal interactions in NPCs but not mESCs (Fig. 6c; for example, 130–160 Mb). Overall, however, virtual 4C data are consistent with a layered organization

of both Xs regardless of cell types, XCI states or SmcHD1 genotypes. The SD domains showed earlier RT than the SI domains (Fig. 6f), which might reflect their distinct compartmentalization, extrapolating from the tight RT-subcompartment relationship[10].

## The outermost SD domains on the Xi are rich in XCI escapees
The XCI escapees are located near the Xi surface[16,39,43,44]. Given the outermost position of the SD layer, we asked how well the SD domains overlap with escapees. Using a comprehensive NPC escapee list[17], the SD domains showed more than 54- and sixfold higher escapee density than the CL and SI domains, respectively (Fig. 7a and Supplementary Fig. 9c), a trend also observed in non-NPC cells[46,47] (Supplementary Fig. 9c,d). Consistently, escapees were significantly enriched in regions with frequent interchromosomal interactions (Fig. 7b). The escapees were positioned on the surface of both Xs based on interchromosomal interaction frequency (Fig. 6a–d), meaning that this is their default positioning regardless of expression states.

To test whether this is conserved in other species, we asked whether human Xi regions with frequent interchromosomal interactions are also escapee rich. We used a published hTERT-RPE1 Hi-C data[18] and performed virtual 4C. Although escapees are not necessarily conserved between mice and humans[1], human escapees were indeed significantly enriched in Xi regions with frequent interchromosomal interactions (Fig. 7c and Supplementary Fig. 9e), suggesting their positioning close to the Xi's outermost surface.

Thus, it is possible that escapees escape XCI because their default positioning on the Xi surface makes their repression state inherently unstable, predisposing them to be easily reactivated. Likewise, SD-domain gene reactivation in SmcHD1-mutant NPCs (Fig. 4b) can be interpreted as preferential derepression of Xi surface domain genes by the absence of SmcHD1. Consistent with this idea, even the nonescapees are reactivated more strongly in SD domains compared to SI and CL domains in SmcHD1-mutant NPCs (Fig. 7d). However, escapees were more strongly reactivated in all domain categories analyzed (Fig. 7d), suggesting that both domain-level and individual gene-level factors contribute to XCI escape.

Last, we asked whether strong gene promoters could drive the characteristic SD-domain positioning. As strong promoters exhibit high CpG density[48,49], we used this as a proxy for strong promoters and analyzed their distribution on the Xi. We observed a similar distribution of high CpG-containing promoters in the SD and SI domains

**Fig. 6 | The SD but not SI domains on the X chromosomes frequently contact other chromosomes in WT and SmcHD1-mutant NSCs. a,b**, Circos plots are shown, which show interactions of 4C-seq viewpoints (red arrowheads) with the rest of the genome in NSCs (**a**) and mESCs (**b**) (representative results from replicate 2). Each line represents a significant interchromosomal interaction. **c**, Using a published allele-specific Hi-C data[11], we created virtual 4C-seq profiles of WT and SmcHD1-mutant (KO) NPCs from 417 viewpoints (VPs) (400-kb bins) and plotted the number of significant interchromosomal interactions for each bin on the cas-Xa and the mus-Xi, as well as the KO–WT differential. Escapee positions are shown[17]. A similar plot was generated for the Xa in mESCs

without allelic separation[11]. Blue lines show *Dxz4* position. **d**, Boxplots showing the number of significant interchromosomal interactions of virtual 4C-seq viewpoints in SD, SI, CL and CE domains on the Xs in WT NPCs and mESCs. *P* values were obtained from a one-sided Wilcoxon signed-rank test with the Bonferroni correction. **e**, Boxplots as in **d** comparing WT versus SmcHD1-KO Xa or Xi in NPCs. *P* values were obtained from one-sided paired Wilcoxon signed-rank test. **f**, Comparison of BrdU-IP RT values of SD and SI domains on the Xi during mESC differentiation. *P* values were obtained from one-sided Wilcoxon signed-rank test. *N*, the number of genomic bins in each class in **d** and **e**.

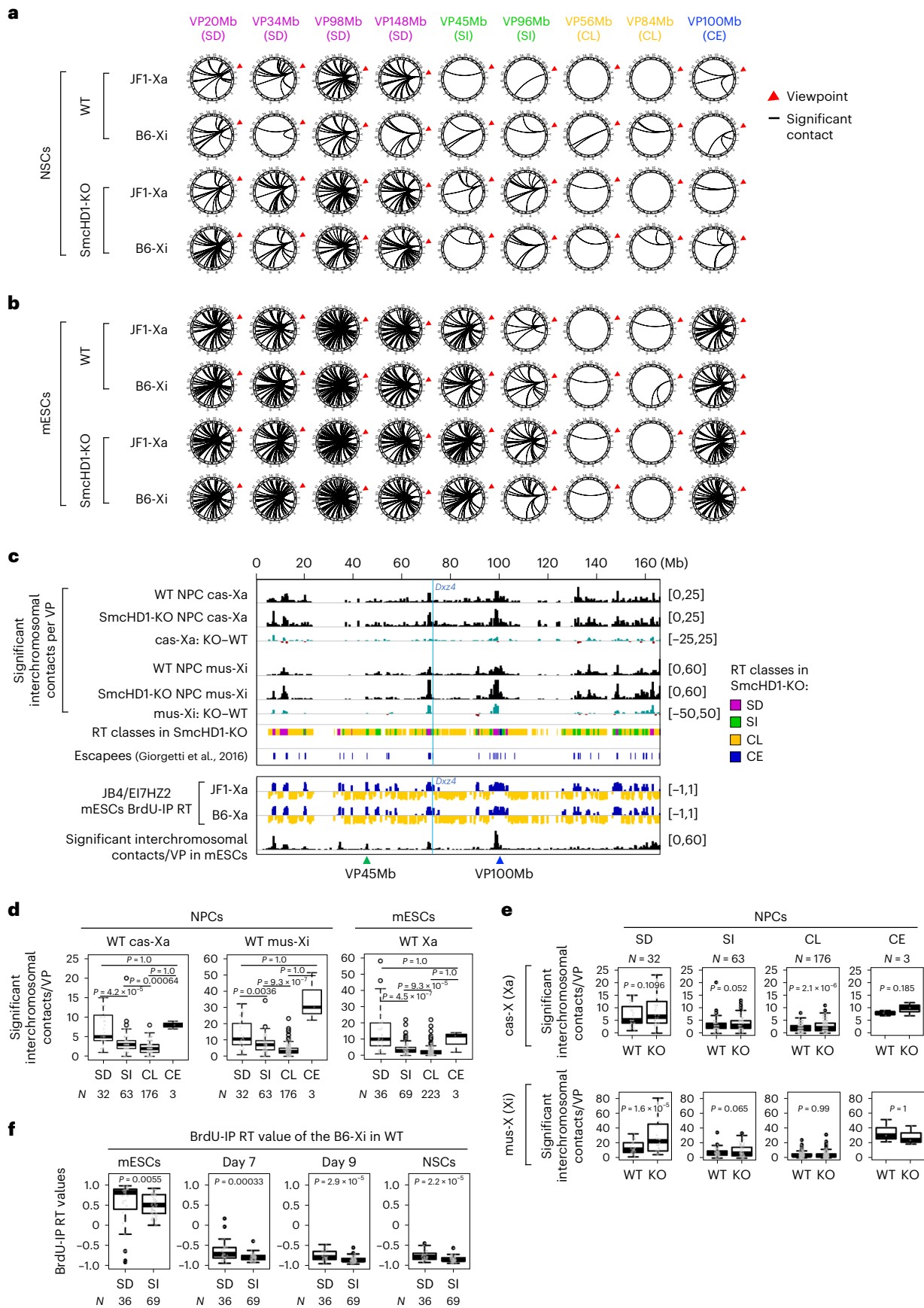

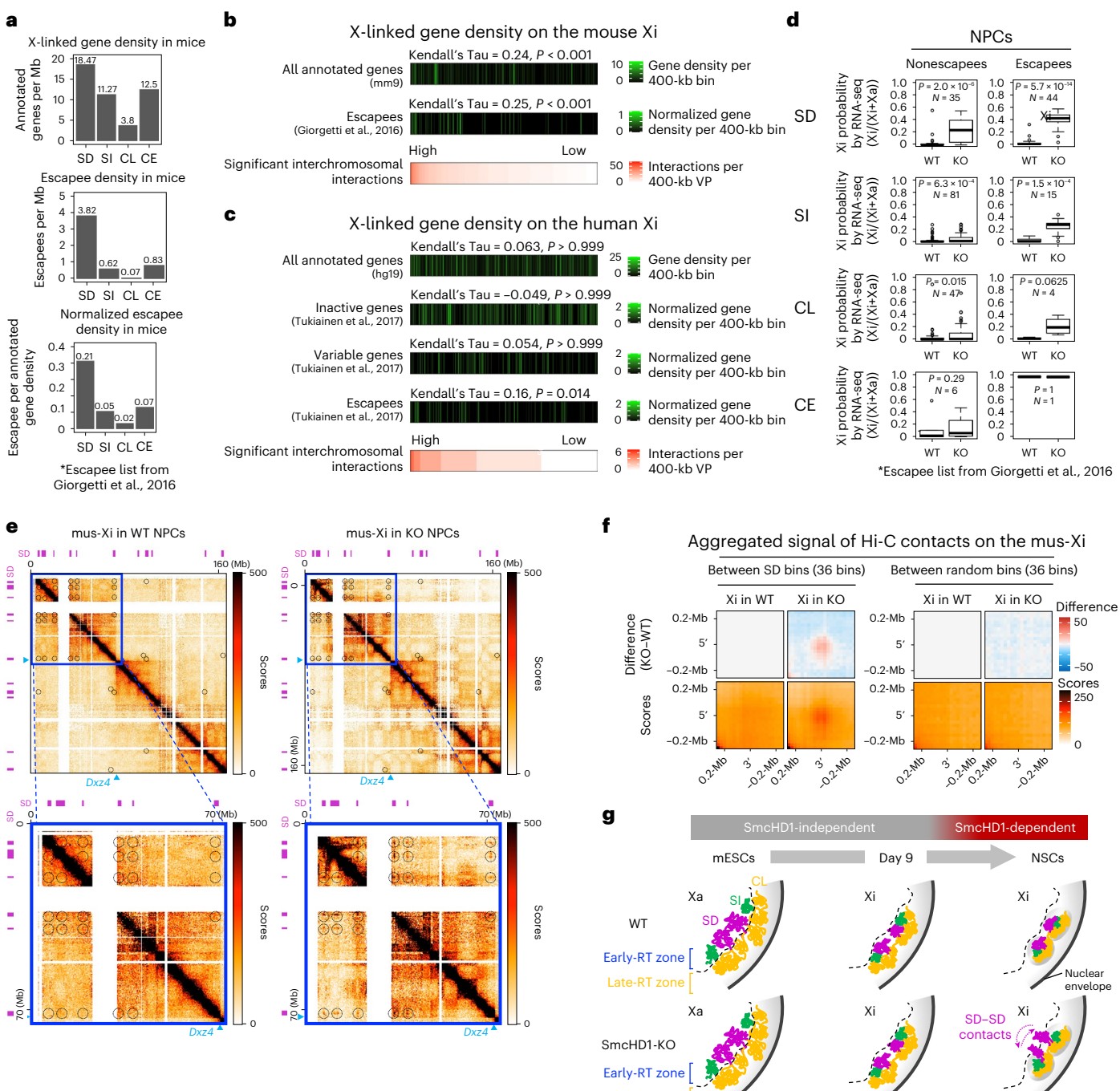

**Fig. 7 | Escapee distribution on the Xi and the identification of SD–SD interactions in SmcHD1-KO NPC Hi-C data. a**, Comparison of gene density on the X among RT classes (genes per Mb). Top, mouse annotated genes (mm9 Ref-seq genes); middle, escapees[17]; bottom, escapees normalized by Ref-seq gene density. **b**, Densities of all annotated Ref-seq genes and escapees[17] normalized by Ref-seq gene density (middle) relative to significant interchromosomal interaction frequency as assayed by virtual 4C of NPCs (bottom)[11]. **c**, Similar plots to **b** were made after virtual 4C (388 viewpoints, 400-kb each) of the human X chromosome using a hTERT-RPE1 Hi-C data[18]. The inactive (genes subject to stable XCI), variable (genes variably escaping from XCI) and escapee genes were defined based on a systematic human transcriptome study[56]. Shown are their densities normalized by UCSC gene density. *P* values in **b** and **c** were obtained from Kendall's rank correlation test with the Bonferroni correction. **d**, Comparison of Xi probability of X-linked gene expression in SmcHD1-mutant (KO) and WT NPCs as in Fig. 4a,b. Genes were classified into nonescapees and escapees[17] and into SD, SI, CL and CE classes. *P* values were obtained from one-

sided paired Wilcoxon signed-rank test. *N*, the number of X-linked genes in each class. **e**, Hi-C heatmaps of the mus-Xi in WT and SmcHD1-mutant (KO) NPCs from Wang et al.[11] (250-kb resolution). Black circles on Hi-C heatmaps indicate strong SD–SD interactions found in SmcHD1-mutant NPCs. The bottom panel shows enlarged views of blue boxes in the top panel. Magenta, SD domains; blue arrows, *Dxz4*. **f**, Aggregated plots of interactions between SD (left) and random bins (right) by Hi-C. **g**, A proposed model for the formation of the multi-layered 3D structural organization of the Xi. During XCI, an EtoL compartment switch of the Xi occurs first (SD and SI domains) and is followed by a further 3D reorganization of the Xi through the actions of factors such as SmcHD1. Without SmcHD1, the Xi fails to maintain its late replication and 3D structure later during differentiation, resulting in frequent protrusion of SD domains located close to the surface of the Xi (but not SI domains), which occasionally interact with each other. Whereas the figure shows the contact between protruded SD domains, it is also possible that SD–SD domain interactions could occur inside the Xi core (Supplementary Text 2).

(Supplementary Fig. 9f), confirming a previous report[50]. Thus, strong promoters cannot account for the SD-domain positioning.

## SD–SD domain interactions are captured by Hi-C

One confounding enigma was that the previous Hi-C did not identify the SD–SD interactions in SmcHD1-mutant NPCs[11]. However, Hi-C does capture what 4C captures, as the virtual 4C data derived from these Hi-C data resembled our actual 4C results and showed SD–SD interactions (Supplementary Fig. 9b). This led us to perform virtual 4C for viewpoints throughout the Xs in NPCs and plot their z scores (Supplementary Fig. 10a–f), which immediately visualized strong SD–SD interactions on the mutant but not WT NPC Xi (Supplementary Fig. 10b,d,f, circled regions). Thus, we revisited the original Hi-C heatmaps[11] and successfully identified the SD–SD interactions in mutant but not WT NPC Xi (Fig. 7e, circled regions). Aggregated plots also revealed strong SD–SD interactions specifically in SmcHD1-mutant NPCs (Fig. 7f).

We also found that the red and blue patterns of the Xa z-score plots resembled the Hi-C principal component 1 (PC1) profile to some extent (Supplementary Fig. 10a,c). This resemblance was also observed on the mutant NPC Xi (Supplementary Fig. 10d), perhaps reflecting the S1 and S2 compartments[11,51]. However, the red and blue patterns on the mutant and WT NPC Xi were similar (Supplementary Fig. 10b,d), suggesting the presence of S1 and S2 compartments on the WT NPC Xi. When we calculated PC1–4 on the Xs and focused on those with more than 8% contribution rates, PC3 of WT NPC Xi resembled the S1 and S2 compartments, which was also true in mouse Patski cells[52] (Supplementary Fig. 10g–i).

Thus, although the Xi is more compact than the Xa, they still seem to share certain structural features. While the megadomain structure becomes prominent as the Xi becomes compact (Supplementary Fig. 10g,j), the S1 and S2 or A and B compartment features still remain on NPC Xi (Supplementary Fig. 10g,h). When the Xi heterochromatin is disturbed by perturbations such as SmcHD1 mutation, the relative contribution of different structural features change, leading to, for instance, the emergence of S1 and S2 compartments as PC1 (ref. 11) or enhanced TAD boundaries in SmcHD1-mutant cells[11,13]. In addition, new structural features, namely protrusion of SD domains and their interactions, emerge on the loss of SmcHD1.

## Discussion

In this study, we used scRepli-seq and analyzed the Xi's RT regulation during mESC differentiation, and explored its compartment organization. Our results demonstrate that (1) the entire Xi is replicated rapidly and uniformly in late S in a given cell but with cell-to-cell RT heterogeneity (Fig. 1e–h); (2) the Xi has a 'layered' organization, which is present already in mESCs, suggesting that this is the default architecture of the X independent of XCI (Fig. 6); (3) SmcHD1 is required for maintaining the XCI state, late-S replication and proper 3D architecture of domains located on the Xi's outermost layer (Figs. 2a–c, 3b and 4–7); (4) the Xi's outermost layer is escapee rich in mice and humans, which becomes preferentially reactivated in SmcHD1-mutant mouse cells (Fig. 7a–d) and (5) the 3D organization defects of the Xi surface domains in SmcHD1-mutant cells can be captured by Hi-C (Fig. 7e,f). Taken together, while the Xi appears to be uniformly compacted in 3D, our results indicated that the positioning relative to the Xi surface can explain regional differences in Xi heterochromatin stability, which was manifested on SmcHD1 mutation or XCI escape (Fig. 7g).

The Xi's chromosome-wide late-S replication has been known since 1960 (ref. 20). However, because roughly 70% of the Xa is late replicating, the EtoL RT switch was assumed to affect only the remaining roughly 30% of the future Xi (Fig. 1d). This was not the case, however, and both RT advances and delays generated a uniformly late-replicating Xi that completed replication within a few hours (Fig. 1e–h), which was consistent with and explain the rapid replication of the mouse C2C12 myoblast Xi in mid- to late S by live imaging[26]. Because developmental

RT changes reflect preceding A and B compartment changes[25], we predict that the Xi's RT changes also reflect its compartment reorganization. Our results validate and update the 'fast and random' Xi replication model[28] and reveal the Xi's chromosome-wide RT reorganization process.

We found relatively large cell-to-cell Xi RT heterogeneity within the second half of S (Fig. 1e–g). Because different Hi-C subcompartments show distinct RT[10], the cell-to-cell RT heterogeneity of the Xi and its RT being out of phase with the rest of the genome likely reflect Xi's compartment states. For instance, the Xi may form its own compartment, which could be slightly variable among cells regarding its position relative to the late-replicating B compartment, resulting in cell-to-cell RT heterogeneity.

The Xi constantly changed its conformation during differentiation, even after becoming late replicating (Fig. 3b and Supplementary Fig. 6). Thus, the Xi may undergo a two-step structural change in which the EtoL domains first alter their compartments in an SmcHD1-independent manner, followed by a process that requires SmcHD1 for further structural reorganization and/or maintenance (Fig. 7g). While the SmcHD1-mutant Xi exhibits pleiotropic defects in gene expression, DNA methylation and histone modifications[14,35–37], recent reports indicate a direct role of SmcHD1 in regulating the higher-order chromosome architecture[11,12]. In particular, SmcHD1 knockout (KO) in cells that have established the Xi clearly showed that the higher-order Xi architecture is altered without gross changes in gene expression[13,15].

SmcHD1 shows binding affinity throughout the Xi[11,13,53] and yet the Xi's RT reversal in SmcHD1-mutant NSCs was specific to the SD domains. Our data indicate that being near the Xi surface makes genes in such regions relatively unstable for maintenance of silencing, possibly because it is easier to make them physically loop out of the Xi chromosome territory than genes underneath the surface layer. Looping out may be a cause or a consequence of gene reactivation. However, being on the X-chromosome surface before XCI cannot be a consequence of reactivation. Therefore, we favor the view that such default 3D architecture of the X forms the basis for regional differences in heterochromatin stability. It has been a longstanding debate whether a given 3D genome architecture is a cause or a consequence of transcription[54]. We believe that our observations represent one of the best examples corroborating the causal role of the 3D genome organization on genome function.

If the default architecture causally affects local differences in Xi heterochromatin stability, it should predefine the sensitivity to perturbations other than SmcHD1 KO. We found that XCI escapees were overrepresented in Xi's outermost surface in mice and human (Fig. 7a–c). By contrast, CL domains rarely contained escapees (Fig. 7a and Supplementary Fig. 9c,d). These observations comprehensively validated the widespread notion that escapees are located near the Xi surface[16,43,44,55].

Escapees were more strongly reactivated than nonescapees in all domain categories analyzed (Fig. 7d), indicating that some intrinsic factors render the escapees more easily reactivated[1,2,5]. However, SD-domain escapees were more strongly reactivated in SmcHD1-mutant NPCs than SI-domain escapees (Fig. 7d). In addition, some SD-domain nonescapees were strongly reactivated in mutant cells, which was much less prominent in SI domains (Fig. 7d). Thus, as the SD domains are closer to the Xi surface than the SI domains, this might make them inherently more unstable. In such ways, SmcHD1-mutant cells can serve as an excellent model to explore XCI escape. Moreover, our work demonstrates the potential functional significance of subcompartments, which is tightly coupled with distinct RT regulation[10].

Interchromosomal interaction analyses by actual and virtual 4C played a critical role in our work. Moreover, we could identify SD–SD interactions in the SmcHD1-mutant NPC Hi-C data (Fig. 7e,f). Thus, routine 4C and Hi-C analyses can overlook important features, and 4C and Hi-C data contain more information to be discovered in the future.

Given the discovery of the potential importance of the default structure of the X, the next challenge would be to decipher the molecular basis of such default 3D genome organization. However, while the interchromosomal interaction patterns were similar between the two Xs in WT mESC and NPCs or SmcHD1-mutant NPCs, there were small differences between mESCs and NPCs (Fig. 6a–c), indicating that the default structure can change during differentiation. In addition, SmcHD1-mutant cells exhibit slightly different RT in different cell types[13], suggesting that SmcHD's contribution to the Xi architecture is cell-type specific. This could be partly explained by cell-type specific Xi compartment organization. It will be important to distinguish the intrinsic ('default') and extrinsic ('epigenetic') aspects of compartment regulation and understand how they are coordinated.

## Online content

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

## Methods

### Cell culture and generation of SmcHD1-mutant female mESCs

The JB4/EI7HZ2 mESCs[32] were grown in 2i/leukemia inhibitory factor (LIF) medium (DMEM (Sigma, D6429-500ML) supplemented with 15% FBS (Gibco 10270-106, lot no. 42Q6272K), 1× pencillin–streptomycin (pen–strep) (Nacalai, 09367-34), 1× nonessential amino acid (Nacalai, 06344-56), 0.1 mM β-mercaptoethanol (Gibco, 21985-023), 1,000 U ml⁻¹ LIF (Nacalai USA, NU0012-2), 1 μM PD0325901 (Wako, 162-25291), 3 μM CHIR99021 (Wako, 034-23103); 2i/LIF refers to MEK and GSK3 inhibitors (2i) and LIF) on culture plates coated with 0.2% gelatin. Medium also contained 0.2 mg ml⁻¹ G418 (InvivoGen, ant-gn-1) and 25 μg ml⁻¹ Zeocin (Invivogen, ant-zn-1) to maintain the XX karyotype. SmcHD1-mutant mESCs were generated from JB4/EI7HZ2 mESCs by CRISPR–Cas9 using the following single guide RNA: GCTGTCGCAGTGGTAGATAA. The following primers were used to genotype SmcHD1-mutant clones: forward (Fw): 5′-TAACTCTGTAGAGCAGGCTG-3′ and reverse (Rv): 5′-TCGCACAGACCTCAGGAAAT-3′. The absence of SmcHD1 protein in SmcHD1-mutant mESCs was confirmed by western blotting using anti-SmcHD1 antibody (1:500 dilution, Sigma HPA039441) and anti-alpha tubulin antibody (1:1,000 dilution, loading control; Abcam ab7291). Epiblast stem cells (EpiSCs) and MEFs were isolated from E6.5 and E12.5 embryos, respectively. The housing conditions for the mice were as follows: they were exposed to a dark/light cycle in darkness from 19:00 to 7:00 and in light from 7:00 to 19:00 (at an intensity of 200 lux). The temperature was maintained at 22 ± 2 °C, and the humidity level was kept at 50 ± 10%. All the experimental procedures using animals were approved by the Institutional Animal Care and Use Committee of Kindai University. EpiSCs were derived in the presence of IWP-2 according to the previous report[57]. MEFs and hTERT-RPE1 cells were cultured in DMEM + 10% FBS and 1× pen–strep.

### Differentiation of mESCs to day 7 or 9 early neurectoderm cells

WT and SmcHD1-mutant JB4/EI7HZ2 mESCs were differentiated to neurectoderm as described in ref. 25. Briefly, JB4/EI7HZ2 mESCs were adapted for two passages (4 days) in N2B27 + 2i/LIF medium (NDiff227 (Cellartis, Y40002) supplemented with 0.1 mM β-mercaptoethanol, 1,000 U ml⁻¹ LIF, 1 μM PD0325901 and 3 μM CHIR99021) on culture plates coated with poly-L-ornithine (0.01% poly-L-ornithine solution; Sigma no. P3655) and 300 ng ml⁻¹ Laminin (Corning, 354232) in DMEM/ F12:Neurobasal medium (1:1) (DMEM/F12 (Nacalai, 11581-15) and neurobasal medium (Gibco, no. 21103-049)). Next, mESCs were dissociated by TrypLE (Gibco, 12604021) and differentiated to epiblast-like cells for 2 days in the presence of Activin A, bFGF and knockout serum replacement (KSR) (NDiff227, 20 ng ml⁻¹ Activin (Shenandoah/Cosmobio SBI 100-43), 12 ng ml⁻¹ FGF2 (Peprotech 100-18B), KSR (Gibco, 10828-028, lot no. 2170051)) on fibronectin coated plates (16.67 ng ml⁻¹ fibronectin (Invitrogen, 33016-015) in 1× PBS (TaKaRa, T900)), collected by trypsinization and switched to aggregation (embryoid bodies (EBs)) culture in Nunclon Sphera 96-well plates (Thermo Fisher Scientific, no. 174925) starting from 4,000 epiblast-like cells per well in GMEM + 15% KSR medium (GMEM (Sigma, G5154-500ML), 15% KSR, 1× pen–strep, 1× nonessential amino acid, 1× sodium pyruvate (Nacalai, 06977-34), 0.09 mM β-mercaptoethanol and 2 mM L-glutamine (Nacalai, 16948-08)). To gather 7- or 9-day differentiated mESCs, aggregates (EBs) were collected and washed with 1× PBS, then treated with 0.25% Trypsin (Nacalai, 32777-44) for 5–10 min at 37 °C to achieve single-cell suspension for further experiments.

### Derivation of NSCs from WT and SmcHD1-mutant mESCs

NSCs were derived from WT and SmcHD1-mutant JB4/EI7HZ2 mESCs and cultured as described[33]. Briefly, mESCs were dissociated by TrypLE and 0.1 × 10⁶ cells were resuspended in 2 ml of NSC differentiation medium (DMEM/F12:neurobasal medium (1:1), 0.5× N₂ supplement (Thermo Fisher, 17502048), 0.5× B27 supplement (Invitrogen, 17504-044), 2 mM L-glutamine, 1× pen–strep, 75 μg ml⁻¹ BSA

(Sigma, A3311-10G), 25 μg ml⁻¹ Insulin (Sigma, I1882-100MG), 0.1 mM β-mercaptoethanol). Then, the cells were seeded on 35-mm culture wells coated with 0.2% gelatin and cultured for 7 days. NSC differentiation medium was changed every 1–2 days. On day 7, differentiated cells were dissociated and 0.5 × 10⁶ cells were resuspended in 2 ml of NSC maintenance medium (DMEM/F12, 1× N₂ supplement, 2 mM L-glutamine, 1× pen–strep, 75 μg ml⁻¹ BSA, 25 μg ml⁻¹ Insulin, 10 ng ml⁻¹ FGF2, 10 ng ml⁻¹ epidermal growth factor (Funakoshi, 2028-EG-200)) and cultured on noncoated petri dishes to induce cell aggregate formation. After 3 days, aggregates were collected by spinning down at 1,000 rpm for 30 s and transferred to new 35-mm culture wells coated with 0.2% gelatin and cultured until the aggregates attached to the surface and produced NSC outgrowth (which typically takes 3–10 days). NSC differentiation medium and NSC maintenance medium also contained 0.2 mg ml⁻¹ G418 and 25 μg ml⁻¹ Zeocin to maintain the XX karyotype. PCR with reverse transcription, Nestin immunostaining (1:200 dilution, Wako, 7A3) and RNA-seq were performed to confirm the identity of NSCs. Primers used for PCR with reverse transcription to confirm the identity of NSCs were as follows: Oct3/4-Fw: 5′-GACAACAATGAGAACCTTCAGG-3′; Oct3/4-Rv: 5′-TGATCTTTTGCCCTTCTGGC-3′; Olig2-Fw: 5′-TTACAGACCGAGCCAA CACC-3′; Olig2-Rv 5′-GGCAGAAAAAGATCATCGGG-3′; Ascl1-Fw: 5′-AGGAACAAGAGCTGCTGGAC-3′; Ascl1-Rv: 5′-TGCAGAGACACTG TTGGAGC-3′. Primers used for PCR with reverse transcription to confirm allele-specific Xist expression were as follows: Fw: 5′-CATCGGGGCTGTGGATACCT-3′; Rv: 5′-AGCACAACCCCGCAAATG CTA-3′. The PCR products amplified by the above primers were subsequently digested with PvuII, whose restriction site is present in the fragment derived from JF1-X but B6-X.

### Xist RNA-FISH and sequential Xist RNA- or DNA-FISH

We followed our standard RNA-FISH and Xist RNA- or DNA-FISH protocols as described in ref. 25. After single-cell suspension with trypsin, cells were incubated in 75 mM KCl hypotonic solution at room temperature for 15 min and fixed in methanol and glacial acetic acid (3:1) at −20 °C for at least 1 h before use. For Xist RNA-FISH, pXist complementary DNA-SS12.9 plasmid was used as a probe template[58]. For SD, SI and CL domain DNA-FISH probes, the following bacterial artificial chromosomes (BACs) were used: RP23-304N5 (68-CL), RP23-131L3 (70-SI), RP23-378I14 (71-SD), RP23-211L1 (94-CL), RP23-470C6 (96-SI), RP23-152F17(98-SD) and RP23-413L19 (148-SD). Briefly, BACs and plasmids were individually labeled with fluorescence-dUTP (Green-dUTP (Enzo Life Sciences no. 02N32-050), Red-dUTP (Enzo Life Sciences no. 02N34-050), or Cyanine 5-dUTP (Perkin Elmer NEL579001EA)) by nick translation (Abbott Molecular no. 07J00-001 (32-801300)). Labeled DNA probes, mouse Cot-1 (Thermo Fisher Scientific no. 18440-016) and salmon sperm DNA (Thermo Fisher Scientific no. 15632-011) were ethanol-precipitated, resuspended in hybridization buffer (10% dextran sulfate, 2× SSC, 1% Tween-20, 50% formamide) and denatured at 80 °C for 10 min before hybridization. For Xist RNA-FISH, cells fixed in methanol and glacial acetic acid (3:1) were dropped onto glass slides and dried for 15 min at room temperature. Slides were washed with 2× SSC and dehydrated in a series of 5-min washes with 70, 90 and 100% ethanol at room temperature. After an overnight hybridization at 37 °C, slides were washed with 2× SSC three times at 45 °C and counterstained with 1 μg ml⁻¹ 4,6-diamidino-2-phenylindole (DAPI) in 2× SSC before mounting with Vectashield (Vector Laboratories no. H1000). For sequential RNA- or DNA-FISH, Xist RNA-FISH was performed first as described above. Xist RNA-FISH signals and their xy coordinates were recorded by DeltaVision Olympus IX71 equipped with Olympus PlanApo ×60 1.42 numerical aperture (NA) oil objective, using the standard SoftWoRx acquisition software (v.6.5.2). For DNA-FISH after Xist RNA-FISH, coverslips were removed after recording by DeltaVision. Slides were washed with 2× SSC three times at 45 °C and incubated in 10 μg ml⁻¹ RNaseA in 2× SSC for 1 h at 37 °C. Slides were washed once

with 2× SSC and dehydrated by sequential 5-min washes with 70, 90 and 100% ethanol at room temperature before being air-dried at 58 °C for 1 h. Slides were then denatured in 70% formamide in 2× SSC at 80 °C for 3 min, dehydrated by sequential washes with cold 70, 90 and 100% ethanol, and were again air-dried at room temperature until hybridization. After an overnight hybridization at 37 °C, slides were washed with 50% formamide in 2× SSC at 45 °C three times, washed again with 0.1% SSC at 60 °C three times and counterstained with 1 μg ml⁻¹ DAPI in 2× SSC before mounting with Vectashield. We recorded DNA-FISH signals by DeltaVision at the same *xy* coordinates from *Xist* RNA-FISH. Images were analyzed using the Fiji software. Because *xy* coordinates had slightly shifted during the experiments, we corrected the positions of *Xist* RNA- and DNA-FISH images by TurboReg macro[59] using DAPI signals of each image as references.

### Imaging analysis of *Xist* RNA- and DNA-FISH signals

The contour of the *Xist* RNA signals (*Xist* cloud) was automatically drawn in each nucleus using a custom Fiji macro. Briefly, an image channel corresponding to *Xist* RNA was subject to 'Enhance Contrast' by saturated at 0.3 with normalization, followed by smoothing through the 'Mean Shift' plugin with spatial value of 10 and color at 25. The image was then converted to binary, and the *Xist* cloud region of interest (ROI) was identified using the 'Analyze Particles' tool, which was defined as the outer edge. The inner edge of the *Xist* cloud was subsequently defined by drawing a scaled ×0.65 ROI. The *xy* positions of DNA-FISH signals were detected using the 'Find Maxima' tool with a prominence threshold of more than 120. Only nuclei with a single *Xist* cloud and two clusters of DNA-FISH signals were analyzed. The DNA-FISH signal localization relative to the *Xist* RNA cloud was manually scored based on four categories: fully overlapped (signal inside the inner edge of the *Xist* cloud), inner edge (signal inside the *Xist* cloud but in between the inner and outer edge), outer edge (signal just outside the *Xist* cloud at its outer edge) and protruded (signal outside the *Xist* cloud). The distance between the DNA-FISH signal relative to *Xist* centroid and between two DNA-FISH signals were calculated based on their *xy* coordinates. The distance was normalized by *Xist* Feret diameter in the same nucleus.

### Sample preparation for RT profiling by BrdU-IP Repli-seq

We followed the BrdU-IP protocol as described[30,60]. Cells or EBs were incubated in a medium containing 10 mM BrdU for 2 h before cell collection. After trypsinization, single-cell suspension was fixed in 75% ethanol. For fluorescence-activated cell-sorting (FACS), we stained fixed cells with Propidium iodide (Nacalai, 29037) and used a Sony SH800 cell sorter (ultra-purity mode) to sort early- and late-S-phase cell population (at least 10,000 cells per fraction). We used a Bioruptor UCD-250 (Sonic Bio) for genomic DNA sonication (high-output mode), with ON/OFF pulse times of 30 s/30 s for 6 min in ice-cold water. BrdU-incorporated DNA was immunoprecipitated using anti-BrdU antibody (BD Biosciences Pharmingen, 555627). After BrdU-IP, immunoprecipitated DNA samples were subject to whole-genome amplification with a SeqPlex kit (Sigma, SEQXE). NGS libraries were constructed from early- and late-replicating DNA after whole-genome amplification with an NGS LTP Library Preparation Kit (KAPA, KK8232) according to the manufacturer's instructions and were subjected to NGS with the HiSeq X Ten system.

### Sample preparation for scRepli-seq profiling

Single cells from the whole-S-phase were sorted with a Sony SH800 cell sorter using single-cell mode. Three gates corresponding to early-, mid- and late-S-phases were set before sorting to select the mode for binarization analysis (Computation associated with the RT profiling of single cells). Sample preparations were performed as described in ref. 31. In total, 120 and 96 single cells for WT and SmcHD1-mutant NSCs (112 S-phase cells and eight G1-phase cells for WT NSCs, and 88 S-phase cells and eight G1-phase cells for SmcHD1-mutant NSCs) were prepared and

analyzed. After filtering out cells with X chromosome abnormalities and abnormal median absolute deviation scores, 95 S-phase cells and four G1-phase cells were subjected to scRepli-seq for WT NSCs, while 85 S-phase cells and four G1-phase cells for SmcHD1-mutant NSCs were subjected to scRepli-seq.

### Allele-specific and nonspecific NGS mapping for RT profiling

Paired-end reads were used as single-end reads. The raw fastq files were trimmed to remove adapter sequences by using the trim_galore v.0.6.6 (−quality 20−phred33−length 35) and cutadapt program v.1.15 before mapping. We performed two-step adapter trimming, first removing the Illumina adapter on the basis of the index of each NGS library and then removing the SEQXE adapter[31]. For mapping to non-haplotype-resolved mouse mm9 reference genome, bwa (v.0.7.17-r1188) was used (command, bwa mem). The Picard tool was used to remove duplicated reads and defined with a MAPQ ≥ 10 as uniquely mapped reads. For haplotype-resolved analysis, we constructed the B6/JF1-specific diploid genome as described in ref. 58, which was used as a reference for the JB4/EI7HZ2 cell line. For haplotype-resolved mapping, bwa (v.0.7.17-r1188) was used (command, bwa aln ≥ bwa samse). Our in-house hTERT-RPE1 phased genome based on SNP information was used as a reference for the hTERT-RPE1 cell line. To obtain the SNP information, haplotype-phasing analysis of the hTERT-RPE1 genome was performed by using a combination of 10X Genomics linked reads, Hi-C data[18] and Strand-seq data[61]. We defined MAPQ ≥ 16 as allele-specific uniquely mapped reads. For reads uniquely mapped to the B6/JF1 diploid genome, we used the liftOver tool (UCSC Genome Browser) to convert the genome coordinates to the mm9. Among the unique reads, we filtered out duplicated reads with the chromosome start position and strand information identical to an existing read. We also filtered out reads that overlapped with the mm9 and hg19 blacklists[31]. Reads per sample are shown in Supplementary Table 2.

### Computation associated with RT profiling of cell populations

We followed our established pipeline for BrdU-IP population RT analysis[30]. For non-haplotype-resolved analysis, after mapping, removing duplicate reads and filtering mm9 blacklist, we counted the reads of early- and late-S-phase BrdU-IP samples in sliding windows of 200- at 80-kb intervals and performed reads per million normalization. Then, the ratio of early-S-phase to total read counts ((early-S reads)/ (early-S reads + late-S reads)) was calculated for each bin, and were further converted to make their distribution to fit within ±1. This value was defined as the BrdU-IP RT score of each bin. We filtered out bins whose total read counts were within the bottom 5% of all bins. For haplotype-resolved RT profiling, we followed the exact same procedures, using nonoverlapping 400-kb bins. All BrdU-IP RT profiles were quantile normalized before downstream analysis. We used published BrdU-IP data of CBMS1 mESCs (GSM2904968 and GSM2904969) and CBMS1 day 7 (GSM2905017 and GSM2905018) from Takahashi et al.[30] for comparison. Hierarchical clustering was done using Ward's method with Euclidean distance in R.

### Computation associated with the RT profiling of single cells

We followed our established pipeline for scRepli-seq RT analysis[31]. First, we analyzed non-haplotype-resolved scRepli-seq data to obtain the percentage replication scores of the whole genome for each single cell in 100-kb nonoverlapping bins. Here, X chromosomes were excluded from the analyses to avoid the bias due to the late replicating Xi. For binarization, different options were applied to each cell depending on their FACS sorting gates (2-HMM option for early-S FACS gate: most. frequent.state = '1-somy'; 2-HMM option for mid-S and late-S FACS gates: most.frequent.state = '2-somy'). To analyze the X chromosomes, we performed haplotype-resolved scRepli-seq as described[30] in 400-kb nonoverlapping bins. Percentage replication scores of scRepli-seq results are shown in Supplementary Table 3. To obtain the single-cell

RT value of a given genomic bin, we first calculated the percentage replication value of each genomic bin (that is, the percentage of cells that have replicated the bin) using non-haplotype-resolved binarized whole-S scRepli-seq data. Then, the genomic bins were subdivided into one-percentile groups based on their percentage replication values, from the earliest (the top one-percentile group) to the latest-replicating group of bins (the bottom one-percentile group). Each of these one-percentile groups has a range of percentage replication values with upper and lower limits (note that the range is variable between different groups) and was assigned an average percentage replication value, which was converted to an S-phase time, that is, the single-cell RT value (0–10 h; with a resolution of 0.1 h), assuming a 10 h S-phase. Therefore, the single-cell RT value of a given one-percentile group represents the average time during the 10 h S-phase when the genomic bins within this group replicate. To obtain the single-cell RT value of each genomic bin on the X chromosomes, we first calculated the percentage replication value of each genomic bin using haplotype-resolved binarized whole-S scRepli-seq data in a way similar to the method described above. Then, a given genomic bin was assigned an single-cell RT value of the one-percentile group with a percentage replication value range that contains that of the given genomic bin.

### Allele-specific 4C-seq experiments

The primary primer sequences for the nine viewpoints on the mouse X chromosome are shown in Supplementary Table 4. These viewpoints were selected based on RT domains containing SNPs that allow allele-specific 4C-seq to be performed, as previously described[24]. The 4C-seq inverse primers were designed to have the first read of the paired-end read (P5) to cover a portion of the viewpoint region (near HindIII) and read into the target ROI. The second read of the pair (P7) was designed to cover a portion of the viewpoint region (near DpnII) containing a SNP to distinguish the homologous chromosomes (Supplementary Fig. 4a,b). The 4C-seq inverse primers were designed for the two-step PCR strategy as described in ref. 40. The primary primers are complementary to the ends of a viewpoint facing outward. These primers contained a portion of the Illumina adapter sequence required for the second PCR step. The second primers are universal primers carrying Illumina indexed adapter sequences (Supplementary Table 4). Thus, they can hybridize to the adapter sequences introduced earlier by the primary primers and amplify the first-round PCR products. This resulted in the complete Illumina adapter sequence for pair-end sequencing. 4C-seq was performed essentially as described[40,45] with modifications below. Briefly, $5–10 × 10^6$ cells were cross-linked with 1% formaldehyde for 10 min at room temperature. Cross-linked cells were lysed and the nuclei were digested with a sino.-base cutter, 400 U HindIII (NEB, R0104T), overnight at 37 °C. Fragmented DNA ends were ligated by 50 U of T4 ligase (Roche/Sigma, no. 79900901, more than or equal to 5 U $μl^{-1}$). Then, the purified DNA was cut again with a four-base cutter, 50 U DpnII (NEB, R0543S), overnight at 37 °C. DNA was ligated again by T4 ligase (Roche/Sigma, no. 79900901, more than or equal to 5 U $μl^{-1}$) overnight at 16 °C to generate 4C-DNA. The 4C-DNA was purified by phenol and chloroform extraction and ethanol precipitation. 4C-seq libraries were first amplified from 800 ng of 4C-DNA per viewpoint with 16 cycles of inverse PCR using primary PCR primer sets. A typical 50 μl PCR reaction was performed with 200 ng 4C-DNA per reaction using Phusion High-Fidelity kit (F-553L). PCR products were purified using 0.8× Agencourt AMPure XP beads (Beckman Coulter, A63881), and eluted in 50 μl eluting buffer (QIAGEN, 19086). Then 5 μl of purified PCR products were used to amplify again with 20 cycles of PCR using secondary PCR primer sets. Final PCR products were cleaned up by the QIAGEN PCR purification kit. Diluted 4C-seq libraries were mixed with other libraries and subjected to paired-end sequencing using the HiSeq X Ten system. We read roughly 4 million reads per sample, but due to the nature of the 4C-seq library, which contained large size products, we usually obtained 1–2 million paired-end reads per sample (Supplementary Table 2).

### Allele-specific 4C-seq data analysis

The 4C-seq reads were first separated specifically to B6 or JF1 based on SNPs in read 2 (from P7) using cutadapt (cutadapt -e 0 –trim-n -g ^(Fw primer sequence) -G ^(Rv primer sequence + SNPs) –no-trim– discard-untrimmed; Supplementary Table 5). Only reads with 0% mismatches to the expected sequence and SNPs were kept. Our 4C-seq libraries had an almost equal fraction of reads between two alleles (roughly 50% each; see Supplementary Tables 1 and 5), indicating an unbiased PCR amplification of libraries. P5 reads of each assigned allele were mapped to mouse mm9 genome as single-end alignment and analyzed using an R pipeline[40] with analysis mode: all, –wSize 201. To analyze significant far-*cis* interactions, we used established R pipelines[45] with parameters -w = 100, -W = 3000 and false discovery rate (FDR) of 0.01. To analyze significant interchromosomal interactions, we used established R pipelines[45] with parameters -w = 500 and FDR = 0.01. Hierarchical clustering was done using Ward's method with Euclidean distance in R.

### Virtual 4C-seq analysis

We used published allele-specific Hi-C data of WT (GSM2667262 and GSM2667264) and SmcHD1-mutant NPCs (GSM2667263 and GSM2667265) from Wang et al.[11] and generated a tag directory from two replicates by HOMER (http://homer.ucsd.edu/homer/interactions/). Virtual 4C-seq was performed using 400-kb viewpoints along the X chromosome (417 viewpoints in total) at 5-kb resolution with a default setting by HOMER (analyzeHiC -res 5000 -vsGenome -4C). Far-*cis* interaction analyses were done using Splinter et al.'s pipeline[45] with modifications. Briefly, normalized reads from virtual 4C-seq analysis (at 5-kb resolution) were made binary and the relative enrichment ($z$ score) was calculated using a sliding window of 50 bins (250 kb) against a background window of 1,200 bins (6 Mb) (as medians of -w = 100 and -W = 3000 in the original pipeline[45] are 194 kb and 6.3 Mb, respectively (mm9)). $Z$ scores of each bin from the far-*cis* interaction analyses were exported and the mean of $z$ scores was calculated using nonoverlapping 400-kb bins. To analyze significant interchromosomal interactions, we used modified R pipelines from Splinter et al.[45] using a window of 250 bins (1.25 Mb) and FDR = 0.01 (as the median of -w = 500 in the original Splinter et al. pipeline is roughly 0.9 Mb (mm9)). For virtual 4C-seq analysis of mESCs, we used Hi-C data from Wang et al. (GSM3036556)[11] but without allelic separation and performed as above. For virtual 4C-seq analysis of hTERT-RPE1, we used raw Hi-C data from Darrow et al. (GSM1847521-GSM1847526)[18] and performed haplotype phasing using our in-house SNP information. Only reads with MAPQ ≥ 10 were used for phasing. To achieve better coverage of Hi-C pairs in each haplotype (called p1 and p2, which represent the Xi and the Xa, respectively), we extracted the following pairs: p1–p1, p1–noSNPs and noSNPs–p1 for p1; p2–p2, p2–noSNPs and noSNPs–p2 for p2. Summary Hi-C files were made based on the phased data and were used to perform virtual 4C-seq as above. In total, 389 viewpoints (400-kb) along the human X chromosomes in hTERT-RPE1 cells were analyzed at 5-kb resolution with default setting by HOMER (analyzeHiC -res 5000 -vsGenome -4C). We found that the X chromosomes in hTERT-RPE1 have a translocation of chromosome 10 (chr10). Therefore, after the analysis of significant interchromosomal interactions using modified R pipelines from Splinter et al.[45] as above, we filtered out any significant interactions of the X chromosome with chr10 before downstream analysis.

### PC analysis and heatmaps of published Hi-C data

We reanalyzed the following published Hi-C data (GSE99991, GSE67516) of WT female mESCs (GSM3036556), day-4 EBs (GSM3036557), day 7 EBs (GSM3036558), WT NPCs (GSM2667262 and GSM2667264), SmcHD1-mutant NPCs (GSM2667263 and GSM2667265) and mouse

Patski cells (GSM2863686)[11,52]. After generating allele-specific Hi-C pairs files including only *cis* chrX reads, we converted the pairs file into .cool format in 500-kb bin matrix with *cis*-balancing using cooler (v.0.8.7)[62] and then performed the PC analysis (A and B compartment calling)[6]. PC1–PC4 were used for downstream analysis. Hi-C data of WT NPCs (GSM2667262 and GSM2667264) and SmcHD1-mutant NPCs (GSM2667263 and GSM2667265) were used to generate heatmaps and aggregation plots at 250-kb resolution using cooler (v.0.8.11)[62] and Genova (v.1.0.1)[63].

### Enrichment of *Xist*, H3K27me3, H3K4me3 and SmcHD1 on the Xs

We used *Xist*-CHART, H3K27me3-ChIP–seq, H3K4me3-ChIP–seq and SmcHD1 Dam ID data of WT and SmcHD1-mutant NPCs from Wang et al.[11]. Briefly, scaled *Xist*-CHART profiles from WT (GSM2667251) and SmcHD1-mutant NPCs (GSM2667254) were used, and the mean of *Xist*-CHART was calculated using nonoverlapping 400-kb bins. Scaled H3K27me3-ChIP–seq profiles from WT (GSM2667232) and SmcHD1-mutant NPCs (GSM2667237) were used and the mean of H3K27me3 enrichment was calculated using nonoverlapping 5-kb bins. Scaled H3K4me3-ChIP–seq profiles from mus-Xi in WT (GSM2667231) and SmcHD1-mutant NPCs (GSM2667236) were used and the mean of H3K4me3 enrichment was calculated using nonoverlapping 400-kb bins. Scaled SmcHD1 Dam ID profiles from WT NPCs (GSM3036552 and GSM3036553) were used and the mean of SmcHD1 enrichment was calculated using nonoverlapping 200-kb bins. For comparison, the $\log_2$ ratio of the enrichment values between WT and SmcHD1-mutant NPCs was calculated for each bin ($\log_2$(SmcHD1-mutant/WT NPCs)). We also used H3K27me3-ChIP–seq and SmcHD1-ChIP–seq of MEFs from Gdula et al.[13]. H3K27me3-ChIP–seq profiles ($\log_2$(IP per input)) from WT (GSM3040189) and SmcHD1-mutant MEFs (GSM3040191) were lifted over to mm9 and the mean of H3K27me3 enrichment was calculated using nonoverlapping 10-kb bins. SmcHD1-ChIP–seq profiles ($\log_2$(IP per input)) from WT MEFs (GSM3040183) were lifted over to mm9 and the mean of SmcHD1 enrichment was calculated using nonoverlapping 200-kb bins.

### Analysis of genes with CpG-containing promoters

A list of genes with different CpG-containing promoters was downloaded from Mikkelsen et al.[48]. The numbers of genes with different CpG-containing promoters were calculated in different RT domains.

### RNA extraction and NGS library preparation

Cells were lysed in TRI Reagent (Molecular Research Center, Inc. catalog TR118). RNA was extracted by Direct-zol RNA miniprep (ZymoResearch, R2050). For RNA-seq ($N = 3$ for each sample), library preparation was performed using 300 ng of total RNA following the standard protocol of Illumina Stranded messenger RNA Prep, Ligation (Illumina, 20040534). Adapter indexes used were IDT for Illumina RNA UD Indexes Set A/B Ligation (Illumina, 20040553/20040554). RNA-seq libraries were sequenced as 80-bp single-end reads by the HiSeq 1500 system.

### RNA-seq analysis

We used published RNA-seq data of CBMS1 mESCs (GSM3127813, GSM3127814), CBMS1 day 7 (GSM3127820, GSM3127821), EpiSCs (GSM3127838, GSM3127839), MEFs (GSM3127841, GSM3127842), WT NPCs (GSM2667220, GSM2667221) and SmcHD1-mutant NPCs (GSM2667222, GSM2667223)[11,25] for comparison to our RNA-seq data. Before mapping, the adapter-sequence trimming and removal of low-quality base reads were performed by trim_galore v.0.6.6 (–quality 20–phred33–length 35). Trimmed fastq files were aligned to the mouse genome (UCSC mm9) by tophat2 v.2.1.1 (ref. 64). We removed ribosomal RNA reads from the mapped reads and excluded reads from the X chromosome for genome-wide non-haplotype-resolved analysis. Filtered mapped reads were quantified against the annotated UCSC transcriptome (mm9) to calculate the fragments per kilobase per million mapped fragments values using the Cuffdiff program of the Cufflinks (v.2.2.1)[65]. CummeRbund (v.2.28.0)[66] was used to plot dendrograms of Jensen–Shannon distances between samples.

### Gene density and escapee analysis

Mouse Ref-seq genes (mm9) were extracted from the Integrative Genomics Viewer (https://software.broadinstitute.org/software/igv/home). Human gene lists (hg19) were downloaded from the UCSC Genome Browser (Track: UCSC genes, table: knownCanonical). Escapee lists for the mouse Xi (mm9) were obtained from three studies[17,46,47]. Only genes that overlapped with our Ref-seq gene list were used (Supplementary Table 7). An escapee list for the human (hg19) Xi was obtained from Tukiainen et al.[56].

### Statistics and reproducibility

Statistical parameters including the statistical tests used, exact values of $N$, the exact values of $P$ and Pearson correlation values ($r$) are reported in the figures, figure legends or associated main texts. Statistical significance is determined by the value of $P < 0.05$ by the indicated tests. Number of replicates is indicated in figure legends. In all the boxplots, the whiskers (lines extending from the box) represent the minimum and maximum values in the dataset, excluding any outliers. The whiskers extend to ±1.5 times the interquartile distance, which is the range between the 25th and 75th percentiles. The horizontal bar at the center of the box represents the median or 50th percentiles of data. The lower and upper bounds of the box indicate the 25th and 75th percentiles, respectively. $N$ is the sample size.

### Reporting summary

Further information on research design is available in the Nature Portfolio Reporting Summary linked to this article.

## Data availability

All RT datasets (BrdU-IP and scRepli-seq), 4C-seq and RNA-seq datasets have been deposited at GEO GSE211574 and are publicly available as of the date of publication.

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

## Acknowledgements

We thank S. Kuraku and members of his laboratory for assistance with RNA-seq and NGS, especially S. Kadota, K. Tatsumi and C. Tanegashima. We also thank A. Oji, S. Takahashi, Y. Kondo and T. Ichinose for technical assistance, D. Meyer and J. Ichinose for help with image analysis and R. Cerbus for comments on the paper. This work was supported by RIKEN BDR intramural grants, RIKEN Pioneering Project 'Genome Building from TADs', JST CREST grant no. JPMJCR20S5, MEXT KAKENHI grant no. 18H05530 and JSPS KAKENHI grant no. 20K20582 to I.H., and RIKEN Incentive Research Project and JSPS KAKENHI grant no. 20K15724 to R.P., Grants-in-Aid for Scientific Research on Innovative Areas from MEXT (grant no. 17H06426) to K.N., MEXT KAKENHI grant nos. 18H05532, 22H05599 and 22H02546 to C.O., and Takeda Science Foundation grant and MEXT KAKENHI grant no. 20H00550 to T.S.

## Author contributions

R.P. and I.H. conceived the project. S.I. and T.S. established mESCs, NSCs, EpiSCs and MEFs. R.P., A.T. and S.I. performed the cell culture, mESC differentiation, sample collection and BrdU-IP. R.P. performed scRepli-seq, 4C-seq, RNA-seq, RNA- and DNA-FISH and bioinformatics analyses. H.M, K.N. and C.O. assembled the hTERT-RPE1 chrX reference sequences based on the SNP information. H.M. established pipelines and performed bioinformatic analyses. R.P. and I.H. wrote the paper with valuable comments from S.I., H.M., K.N., C.O. and T.S.

## Competing interests

The authors declare no competing interests.

## Additional information

**Correspondence and requests for materials** should be addressed to Ichiro Hiratani.

# Reporting Summary

## Statistics

For all statistical analyses, confirm that the following items are present in the figure legend, table legend, main text, or Methods section.

| n/a | Confirmed | |
|---|---|---|
| ☐ | ☒ | The exact sample size (*n*) for each experimental group/condition, given as a discrete number and unit of measurement |
| ☐ | ☒ | A statement on whether measurements were taken from distinct samples or whether the same sample was measured repeatedly |
| ☐ | ☒ | The statistical test(s) used AND whether they are one- or two-sided<br>*Only common tests should be described solely by name; describe more complex techniques in the Methods section.* |
| ☒ | ☐ | A description of all covariates tested |
| ☐ | ☒ | A description of any assumptions or corrections, such as tests of normality and adjustment for multiple comparisons |
| ☐ | ☒ | A full description of the statistical parameters including central tendency (e.g. means) or other basic estimates (e.g. regression coefficient) AND variation (e.g. standard deviation) or associated estimates of uncertainty (e.g. confidence intervals) |
| ☐ | ☒ | For null hypothesis testing, the test statistic (e.g. *F*, *t*, *r*) with confidence intervals, effect sizes, degrees of freedom and *P* value noted<br>*Give P values as exact values whenever suitable.* |
| ☒ | ☐ | For Bayesian analysis, information on the choice of priors and Markov chain Monte Carlo settings |
| ☒ | ☐ | For hierarchical and complex designs, identification of the appropriate level for tests and full reporting of outcomes |
| ☐ | ☒ | Estimates of effect sizes (e.g. Cohen's *d*, Pearson's *r*), indicating how they were calculated |

*Our web collection on statistics for biologists contains articles on many of the points above.*

## Software and code

Policy information about availability of computer code

| Data collection | Images were collected by DeltaVision Olympus IX71 inverted microscope and a standard SoftWoRx acquisition software version 6.5.2. |
|---|---|
| Data analysis | We used custom codes from Takahashi et al., Nature Genetics 2019 and Miura et al., Nature Protocols 2020 for BrdU-IP Repli-seq and single-cell Repli-seq (scRepli-seq) analyses. These custom codes are available at https://github.com/kuzobuta/hic_paper_NG_2019.<br>We used R pipelines from Krijger et al., Methods 2020 for 4C-seq analyses.<br>We used R pipelines from Splinter et al, Methods 2012 to analyze significant cis- and trans-interactions of 4C-seq data.<br>We used tophat2 version 2.1.1, (Kim et al., 2013), Cufflinks (Trapnell et al., 2010) and CummeRbund (Trapnell et al., 2012) for RNA-seq analyses.<br>We used Fiji (Schindelin, J et al. 2012) for imaging analyses. We used TurboReg plugin (Thévenaz et al., 1998) for coordinate correction of images. A standard Find Maxima and Mean Shift plugins in Fiji were used.<br>We used HOMER (Heinz et al., 2010) to generate virtual 4C-seq profiles from Hi-C data.<br>We used cooler (Abdennur et al., 2020) for Hi-C analyses and Genova (Van Der Weide et al., 2021) to plot Hi-C heatmaps.<br>We also used the following programs for NGS analyses: trim_galore version 0.6.6, BWA version v.0.7.17-r1188 (Li H et al., 2010), SAMtools (Li et al., 2009), BEDTools (Quinlan and Hall, 2010), Cutadapt (M Martin, 2010), FastQC (http://www.bioinformatics.babraham.ac.uk /projects/fastqc/), Picard (http://broadinstitute.github.io/picard), Liftover (http://hgdownload.soe.ucsc.edu/admin/exe/), AneuFinder (R package, 1.2.1). Cell Sorter Software version 2.1.6 was used for SONY SH800 flow cytometry. |

For manuscripts utilizing custom algorithms or software that are central to the research but not yet described in published literature, software must be made available to editors and reviewers. We strongly encourage code deposition in a community repository (e.g. GitHub). See the Nature Portfolio guidelines for submitting code & software for further information.

## Data

Policy information about availability of data

All manuscripts must include a data availability statement. This statement should provide the following information, where applicable:
- Accession codes, unique identifiers, or web links for publicly available datasets
- A description of any restrictions on data availability
- For clinical datasets or third party data, please ensure that the statement adheres to our policy

All replication timing datasets (BrdU-IP and scRepli-seq), 4C-seq, and RNA-seq datasets have been deposited in Gene Expression Omnibus under accession GSE211574.

## Human research participants

Policy information about studies involving human research participants and Sex and Gender in Research.

| | |
|---|---|
| Reporting on sex and gender | Human research is not involved in this study. |
| Population characteristics | Human research is not involved in this study. |
| Recruitment | Human research is not involved in this study. |
| Ethics oversight | Human research is not involved in this study. |

Note that full information on the approval of the study protocol must also be provided in the manuscript.

# Field-specific reporting

Please select the one below that is the best fit for your research. If you are not sure, read the appropriate sections before making your selection.

☒ Life sciences ☐ Behavioural & social sciences ☐ Ecological, evolutionary & environmental sciences

For a reference copy of the document with all sections, see nature.com/documents/nr-reporting-summary-flat.pdf

# Life sciences study design

All studies must disclose on these points even when the disclosure is negative.

| | |
|---|---|
| Sample size | No sample-size calculation was performed. The sample size was chosen based on previous experience and standards in the field. For scRepli-seq analysis, we selected a sample size that ensures capturing cells at various stages throughout the S phase (Takahashi et al., Nature Genetics 2019). All sample sizes are shown in the figures or legends. |
| Data exclusions | For scRepli-seq analyses, we treated the data as described in Miura et al., Nature Protocols 2020. From a total of 120 single-cell data sets of wild-type NSCs, we excluded 15 that did not pass the 'MAD score' QC, 1 that showed problematic replication timing (RT) data distribution by the manhattan distance, as well as 6 with X chromosome abnormalities and used the remaining 98 data sets for downstream scRepli-seq analysis (4 G1-phase cells and 94 S-phase cells). From a total of 96 single-cell data sets of SmcHD1 mutant NSCs, we excluded 7 that did not pass the 'MAD score' QC and used the remaining 89 data sets for downstream scRepli-seq analysis (4 G1-phase cells and 85 S-phase cells).

For Xist RNA-FISH quantification, we excluded cells that were unclassified from the analysis (which were less than 1% of total cell analyzed). For a sequential Xist RNA-FISH and DNA-FISH experiment, we first excluded cells that did not show signal from Xist or show more than one Xist cloud/nucleus based on RNA-FISH results from the analyses. Next, we excluded cells that showed more than two spots from either SD, SI, or CL DNA-FISH signals near the Xi from the analyses. |
| Replication | All experiments, except for scRepli-seq and RT-PCR, were replicated at least twice, and in all cases, data sets exhibited high correlation among the replicates, assuring data reproducibility. Biological replicates represented by cells from different passages and different collected date. |
| Randomization | Randomization is not relevant to this study because no comparisons between experimental groups were made. |
| Blinding | Blinding was not relevant to this study because all metrics were derived from absolute quantitative methods without human subjectivity. |

# Reporting for specific materials, systems and methods

We require information from authors about some types of materials, experimental systems and methods used in many studies. Here, indicate whether each material, system or method listed is relevant to your study. If you are not sure if a list item applies to your research, read the appropriate section before selecting a response.

## Materials & experimental systems

| n/a | Involved in the study |
|---|---|
| ☐ | ☒ Antibodies |
| ☐ | ☒ Eukaryotic cell lines |
| ☒ | ☐ Palaeontology and archaeology |
| ☐ | ☒ Animals and other organisms |
| ☒ | ☐ Clinical data |
| ☒ | ☐ Dual use research of concern |

## Methods

| n/a | Involved in the study |
|---|---|
| ☒ | ☐ ChIP-seq |
| ☐ | ☒ Flow cytometry |
| ☒ | ☐ MRI-based neuroimaging |

## Antibodies

| Antibodies used | Primary antibody:<br>1. Anti-BrdU antibody for BrdU-IP Repli-seq profiles (dilution to 12.5 ug/ml by PBS) from BD Biosciences Pharmingen (cat. 555627)<br>2. Anti-SmcHD1 antibody for western blot (1:500 dilution) from Sigma (cat. HPA039441; lot K106669)<br>3. Anti-alpha tubulin antibody for western blot (1:1000 dilution) from Abcam (cat. ab7291; lot GR138941-3)<br>4. Anti-Nestin antibody for immunostaining (1:200 dilution) from Wako (cat. 7A3) |
|---|---|
| Validation | All antibodies are commercially available and have associated datasheets from the supplier.<br>1. Anti-BrdU antibody was validated to immunoprecipitate BrdU-containing DNA and used for the previous study (Takahashi et al., Nature Genetics 2019)<br>2. Anti-SmcHD1 antibody was validated by western blot (https://www.sigmaaldrich.com/JP/ja/product/sigma/hpa039441)<br>3. Anti-alpha tubulin antibody was validated by western blot (https://www.abcam.com/products/primary-antibodies/alpha-tubulin-antibody-dm1a-loading-control-ab7291.html)<br>4. Anti-Nestin antibody was validated by immunostaining (https://labchem-wako.fujifilm.com/jp/product/detail/W01W0101-2684.html) |

## Eukaryotic cell lines

Policy information about cell lines and Sex and Gender in Research

| Cell line source(s) | The JB4/EI7HZ2 mESC line (Matsuura et al., Front. Cell Dev. Biol. 2021) originated from Prof. Takashi Sado's laboratory at Kindai University, Japan. The human TERT-RPE1 cell line (Clontech, C4001-1) was obtained from Prof. Chikashi Obuse's laboratory at Osaka University, Japan. |
|---|---|
| Authentication | Karyotype (JB4/EI7HZ2) and SNPs (JB4) were verified using an NGS technology. SmcHD1 mutant JB4/EI7HZ2 mESC line generated in this study was validated by PCR, sequencing, and western blot. Differentiated wild-type and SmcHD1 mutant JB4/EI7HZ2 NSCs generated in this study were validated by RT-PCR, immunostaining, and RNA-seq. |
| Mycoplasma contamination | The cell lines were not tested for mycoplasma contamination. |
| Commonly misidentified lines (See ICLAC register) | No commonly misidentified cell lines were used. |

## Animals and other research organisms

Policy information about studies involving animals; ARRIVE guidelines recommended for reporting animal research, and Sex and Gender in Research

| Laboratory animals | EpiSCs and MEFs were isolated from E6.5 and E12.5 mouse embryos, respectively. EpiSCs were derived in the presence of IWP-2 according to Sugimoto et al., Stem Cell Reports 2015.<br>The housing conditions for the mice are as follows: they are exposed to a dark/light cycle with darkness from 19:00 to 7:00 and light from 7:00 to 19:00 (at an intensity of 200 lux). The temperature is maintained at 22±2 °C, and the humidity level is kept at 50±10%. |
|---|---|
| Wild animals | The study did not involve wild animals. |
| Reporting on sex | Female. |
| Field-collected samples | The study did not involve samples collected from the filed. |
| Ethics oversight | The animals were housed in environmentally controlled rooms, and all the experimental procedures using animals were approved by the Institutional Animal Care and Use Committee of Kindai University. |

Note that full information on the approval of the study protocol must also be provided in the manuscript.

# Flow Cytometry

## Plots

Confirm that:

☒ The axis labels state the marker and fluorochrome used (e.g. CD4-FITC).

☒ The axis scales are clearly visible. Include numbers along axes only for bottom left plot of group (a 'group' is an analysis of identical markers).

☒ All plots are contour plots with outliers or pseudocolor plots.

☒ A numerical value for number of cells or percentage (with statistics) is provided.

## Methodology

| | |
|---|---|
| Sample preparation | Cells were fixed with 75% ethanol, and then stained with propidium iodide (PI) to assess the DNA content of the cells as previously described (Takahashi et al., Nature Genetics 2019). DNA content was analyzed using Sony SH800 cell sorter. |
| Instrument | Sony SH800 Cell sorter (SONY). |
| Software | The data was collected by Cell Sorter Software version 2.1.6. |
| Cell population abundance | 1,000,000-2,000,000 cells were acquired for each sample. Around 90% of the cells showed the typical cell cycle profile. |
| Gating strategy | An FSC/BSC gate was used for gating the population of cells to exclude cell debris. Then, a PI gate was used to exclude doublet cells. The gates for sorting the G1 or desired S-phase fractions were defined on the PI histogram. For BrdU-IP, cells were sorted with the purity mode of the SH800 Cell sorter. For scRepli-seq, cells were sorted using the single-cell mode. |

☒ Tick this box to confirm that a figure exemplifying the gating strategy is provided in the Supplementary Information.

