## [Peer Review File · Nature Structural & Molecular Biology]

Peer Review Information

Manuscript Title: Replication dynamics identifies the folding principles of the inactive X chromosome

Corresponding author name(s): Ichiro Hiratani

Reviewer Comments & Decisions:

Decision Letter, initial version:
--

Message: 17th Jan 2023

Dear Dr. Hiratani,

Thank you again for submitting your manuscript "Replication dynamics identifies the folding principles of the inactive X chromosome". I apologize for the delay in responding, which resulted from the difficulty in obtaining suitable referee reports, together with a significantly reduced editorial capacity over the holidays. Nevertheless, we now have comments (below) from the 2 reviewers who evaluated your paper. In light of those reports, we remain interested in your study and would like to see your response to the comments of the referees, in the form of a revised manuscript.

You will see that while both reviewers are positive about the impact of this study, Reviewer #1 suggests to analyse the cis-elements potentially driving the organization of a sub-compartment in the X chromosome, and Reviewer #2 asks for orthogonal validation of the layered chromosome structure of the Xi for example by DNA-FISH. Please be sure to address/respond to all concerns of the referees in full in a point-by-point response and highlight all changes in the revised manuscript text file. If you have comments that are intended for editors only, please include those in a separate cover letter.

We expect to see your revised manuscript within 6 weeks. If you cannot send it within this time, please contact us to discuss an extension; we would still consider your revision, provided that no similar work has been accepted for publication at NSMB or published elsewhere.

Reporting Summary:

Please note that all key data shown in the main figures as cropped gels or blots should be presented in uncropped form, with molecular weight markers. These data can be aggregated into a single supplementary figure item. While these data can be displayed in a relatively informal style, they must refer back to the relevant figures. These data should be submitted with the final revision, as source data, prior to acceptance, but you may want to start putting it together at this point.

Data availability: this journal strongly supports public availability of data. All data used in accepted papers should be available via a public data repository, or alternatively, as Supplementary Information. If data can only be shared on request, please explain why in

your Data Availability Statement, and also in the correspondence with your editor. Please note that for some data types, deposition in a public repository is mandatory - more information on our data deposition policies and available repositories can be found below: <https://www.nature.com/nature-research/editorial-policies/reporting-standards#availability-of-data>

[Redacted]

Sincerely,
Sara

Sara Osman, Ph.D.
Associate Editor
Nature Structural & Molecular Biology

Referee expertise:

Referee #1: DNA replication, genome organisation

Referee #2: X-chromosome inactivation, genome organisation

Reviewers' Comments:

Reviewer #1:

Remarks to the Author:

The paper by Poonperm et al focuses on deciphering how the inactive X chromosome is progressively structured during the differentiation of mESCs into NPCs. Several properties are studied using state-of-the-art methodologies including single cell replication program (scRepli-seq), 4C, Hi-C haplotype specific analyses. These high quality assays allow significant progress in the understanding of the mechanisms involved in X chromosome inactivation, a question that animates a large community. Moreover, the results obtained on the replication program are of interest to the community working on the regulation of replication.

The key observations are the following:

1) The entire Xi is replicated rapidly and uniformly in late-S in a given cell but with cell-to-cell RT heterogeneity. Although late-S replication of Xi has been known for a long time, this study shows for the first time that the uniform pattern of DNA replication results from both advances and delays of several regions of the chromosome. The cell-to-cell RT heterogeneity explains the previously observed 'fast and random' Xi replication model and reveals a specific mode of replication. The clear connection with 3D organization suggest a model in which the Xi may form its own compartment. The peculiar mode of replication of the Xi chromosome would need further investigation. However, this information would be of most interest to the small community working on replication timing. Furthermore, a detailed understanding of these mechanisms may require a complex set of approaches. It seems to me that the scope of the new information is sufficient in the context of this paper.

2) The Xi has a layered organization that is already present in the Xa in mESC. This suggests that there is an organization probably dictated by information coded by the genome.

Several clues suggest that cis-elements found in strong promoters might be important in establishing the outermost layer of the Xi. Indeed these regions are rich in escapees which were preferentially reactivated in SmcHD1-mutant cells in mice and escapees contain strong promoters. A high density of genes containing strong promoters could be the signal that allows the establishment of this specific sub-compartment of the X chromosome. This hypothesis might be explored and discussed in the paper.

3) SmcHD1 is required for maintaining the XCI state, late-S replication and proper 3D architecture of domains on the Xi's outermost surface layer.

I do not see any major problems with the results, the way they are presented and their interpretation. The conclusions are well demonstrated by the experiments described in the article. Moreover, the clear presentation of a rather complex data set ensures a relatively easy reading. The only frustration remains perhaps a lack of further investigation of potential key genomic cis-information of these new regions (SD) and even SI regions. It is unclear whether the authors have looked for the presence of enriched motifs in these regions compared to the rest of the X chromosome. If this is the case and the search was unsuccessful, it would be interesting to have the information in the article. Such analysis

would allow progress in deciphering the molecular basis of the default 3D organization of the X chromosome.

For all these reasons, it seems to me that the article is appropriate for publication in the NSMB journal with a minor revision including an analysis of the cis-elements potentially driving the organization of a sub-compartment in the X chromosome.

1. Koren, A., and McCarroll, S.A. (2014). Random replication of the inactive X chromosome. *Genome Res.* 24, 64–69. 10.1101/gr.161828.113.

Reviewer #2:

Remarks to the Author:

In female mammalian cells, one of the two X chromosomes forms a stable heterochromatic structure in early development to compensate for the double genetic dose. In this study, the authors employ F1-hybrid mouse embryonic stem cells (mESCs) that differentiate into neural stem cells (NSCs) as a model system to investigate compartment formation during development. Through genome-wide and single-cell Repli-seq, they demonstrate the early-to-late replication timing (RT) switch during X chromosome inactivation (XCI) with high cell-to-cell variability. By using a knockout cell line of the non-canonical SMC family protein SmcHD1, they identify so-called SD domains on the X chromosome, which depend on SmcHD1 for late replication timing and heterochromatic stability. Using 4C-seq and Hi-C, the authors show that these SD regions are in close proximity to other SD regions and autosomes and are enriched for escape genes. The authors propose a transcription-independent hierarchical structure of the inactive X chromosome, with SD elements located on the surface of the Xi core. This manuscript is a valuable study with appropriate length to elucidate the different regulatory layers that control chromosome conformation during X chromosome silencing.

This study is a valuable resource for the interplay between 3D genome organization and heterochromatin stability. It uses an elegant system of JB4/EI7H22 mESCs to ensure 100% skewed XCI and shows the advances and delays in replication timing and cell-to-cell heterogeneity at high resolution through the application of scRepli-seq. The interchromosomal interactions of the individual viewpoints in Figure 5a and b are an interesting feature of the study and convincingly show the increased interaction of SD domains throughout development. We would however propose to validate the main findings with an orthogonal approach. We overall support publication of the study, if the follow points are addressed.

Major Comments

1. One of the major findings of the study is the layered chromosome structure of the Xi. Although the analysis of interchromosomal contacts is an innovative and elegant approach, it would be useful to support the conclusion with an orthogonal method. The subcompartment organization could for example be assayed by DNA-FISH. Visualizing multiple SD domains in the same cell could confirm the interaction of SD domains and their collective protrusion out of the Xi core. Moreover, simultaneous detection of SD and SI probes might support the hypothesis of SI domains being located "just underneath the SD domain layer in NSCs" (line 297). This holds especially true for the VP45Mb viewpoint, which is suggested to be located right below the SD layer. The additional microscopic visualization might increase the understanding of SD domain protrusion.

2. The authors suggest that the susceptibility of the SD domains to SmcHD1 KO is due to the spatial position of these domains at the edge of the chromosome territory and not due to preferential binding of SmcHD1 to these regions. To further support this hypothesis it would be helpful to see the SmcHD1 binding pattern in SD compared to SI domains by e.g. using (published) ChIP-seq or DamID data.

3. A previous study has identified regions on the Xi that are depleted for H3K27me3 in SmcHD1 KO cells (<https://doi.org/10.1038/s41467-018-07907-2>). Are these regions identical with SD domains?

4. In Figure 3b, 4C-seq data from the Xi in NSCs on Day 9 shows increased contacts between the SD regions and weaker cis contacts compared to the other non-SD viewpoints. As the profiles are difficult to compare by eye, they should be quantified and analyzed statistically.

5. The last section of the results is difficult to follow. It could improve the logical flow of the paper to integrate the Hi-C data (Fig. 6e-j) with the 4C-seq data in Figure 3, since both show SD-SD domain interactions. Here, a quantitative analysis of the SD domains identified by 4C-seq vs Hi-C could be useful.

6. As mentioned in the introduction, the DXZ4 repeat region plays a structural key function on the inactive X chromosome. Therefore, it would be helpful to add the location of DXZ4 in the figures to assess its RT class and interaction pattern.

7. SmcHD1 is a well-studied protein which has been shown to play a role in X inactivation and higher-order chromatin architecture. It would be helpful to introduce its known functions from previous research already in the introduction, and also explain S1/S2 compartments.

8. In Fig. 2c, the number of domains in each category should be indicated.

9. We could not access the GEO repository.

Minor Comments

10. In the model proposed in the study, SD domains protrude out of the Xi core, but at the same time cluster together. Does this happen in the same cell or in different cell subsets? It would be helpful to discuss in some more detail how this could be envisioned.

11. In Fig. 5d, CE regions show the highest interchromosomal contacts (higher than SD domains). It would be useful to provide some more information on these regions and on how they differ from SD domains. What factors might keep them in the early replication phase?

12. In Figure 1, it would help the reader's understanding to add a schematic of the Repli-seq method.

13. It would be useful to add the number of EtoL, LtoE, EtoE and LtoL domains and what percentage of the X chromosome they cover to Fig. 1.

14. The term “% replication score” does not read well (line 136).
15. The RT classes abbreviations should be explained in the legend of Figure 2, rather than Figure 3a.
16. It should be indicated in the figure legend of Fig. 6e-j what the grey areas represent.

Author Rebuttal to Initial comments

Point-by-point response to all of the reviewers’ criticisms (reviewer comments in bold italic):

Reviewer #1:

Remarks to the Author:

The paper by Poonperm et al focuses on deciphering how the inactive X chromosome is progressively structured during the differentiation of mESCs into NPCs. Several properties are studied using state-of-the-art methodologies including single cell replication program (scRepli-seq), 4C, Hi-C haplotype specific analyses. These high quality assays allow significant progress in the understanding of the mechanisms involved in X chromosome inactivation, a question that animates a large community. Moreover, the results obtained on the replication program are of interest to the community working on the regulation of replication.

We thank the reviewer for the positive comments and for understanding the value of our work.

Several clues suggest that cis-elements found in strong promoters might be important in establishing the outermost layer of the Xi. Indeed these regions are rich in escapees which were preferentially reactivated in SmcHD1-mutant cells in mice and escapees contain strong promoters. A high density of genes containing strong promoters could be the signal that allows the establishment of this specific sub-compartment of the X chromosome. This hypothesis might be explored and discussed in the paper.

3) SmcHD1 is required for maintaining the XCI state, late-S replication and proper 3D architecture of domains on the Xi’s outermost surface layer.

I do not see any major problems with the results, the way they are presented and their interpretation. The conclusions are well demonstrated by the experiments described in the article. Moreover, the clear presentation of a rather complex data set ensures a relatively easy reading. The only frustration remains perhaps a lack of further investigation of potential key genomic cis-information of these new regions (SD) and even SI regions. It is unclear whether the authors have looked for the presence of enriched motifs in these regions compared to the rest of the X chromosome. If this is the case and the search was unsuccessful, it would be interesting to have the information in the article. Such analysis would allow progress in deciphering the molecular basis of the default 3D organization of the X chromosome.

The reviewer had just one request. The reviewer hypothesized that *a high density of genes containing strong promoters could be the signal that allows the establishment of this specific sub-compartment of the X*

chromosome (i.e., SD subcompartment) and suggested testing this hypothesis by looking for the presence of enriched motifs in the SD regions compared to the rest of the X chromosome.

To test this hypothesis, we investigated whether the SD domains contained a higher number of genes with high CpG density promoters (Mikkelsen et al., Nature 2007), a property associated with strong promoters. We analyzed the percentages of high, intermediate, and low CpG-density promoters in SD, SI, CL, and CE domains. However, we observed a similar distribution of genes with high CpG-containing promoters in both the SD and the SI domains (Supplementary Fig. 9d). These results indicate that strong promoters alone cannot account for this organization, and we discussed this in the main text (page 14).

Reviewer #2:

Remarks to the Author

In female mammalian cells, one of the two X chromosomes forms a stable heterochromatic structure in early development to compensate for the double genetic dose. In this study, the authors employ F1-hybrid mouse embryonic stem cells (mESCs) that differentiate into neural stem cells (NSCs) as a model system to investigate compartment formation during development. Through genome-wide and single-cell Repli-seq, they demonstrate the early-to-late replication timing (RT) switch during X chromosome inactivation (XCI) with high cell-to-cell variability. By using a knockout cell line of the non-canonical SMC family protein SmcHD1, they identify so-called SD domains on the X chromosome, which depend on SmcHD1 for late replication timing and heterochromatic stability. Using 4C-seq and Hi-C, the authors show that these SD regions are in close proximity to other SD regions and autosomes and are enriched for escape genes. The authors propose a transcription-independent hierarchical structure of the inactive X chromosome, with SD elements located on the surface of the Xi core. This manuscript is a valuable study with appropriate length to elucidate the different regulatory layers that control chromosome conformation during X chromosome silencing.

This study is a valuable resource for the interplay between 3D genome organization and heterochromatin stability. It uses an elegant system of JB4/EI7HZ2 mESCs to ensure 100% skewed XCI and shows the advances and delays in replication timing and cell-to-cell heterogeneity at high resolution through the application of scRepli-seq. The interchromosomal interactions of the individual viewpoints in Figure 5a and b are an interesting feature of the study and convincingly show the increased interaction of SD domains throughout development. We would however propose to validate the main findings with an orthogonal approach. We overall support publication of the study, if the follow points are addressed.

We sincerely thank the reviewer for the positive comments and constructive criticisms. We also felt the need to validate the main findings with an orthogonal approach, and the other comments were also very helpful. We hope that we have addressed the points, which will be described below.

Major Comments

1. One of the major findings of the study is the layered chromosome structure of the Xi. Although the analysis of interchromosomal contacts is an innovative and elegant approach, it would be useful to support the conclusion with an orthogonal method. The subcompartment organization could for example be assayed by DNA-FISH. Visualizing multiple SD domains in the same cell could confirm the interaction of SD domains and their collective protrusion out of the Xi core. Moreover, simultaneous detection of SD and SI

probes might support the hypothesis of SI domains being located “just underneath the SD domain layer in NSCs” (line 297). This holds especially true for the VP45Mb viewpoint, which is suggested to be located right below the SD layer. The additional microscopic visualization might increase the understanding of SD domain protrusion.

Based on the reviewer’s suggestion, we performed additional FISH and analyzed the relative positions of the RT domains. Overall, our FISH data validated the 4C data, and we described the data as follows on pages 10-11:

To validate the *Xist* RNA binding data, we performed a sequential *Xist* RNA fluorescent *in situ* hybridization (RNA-FISH) and DNA-FISH using two sets of probes targeting neighboring SD/SI/CL domains in wild-type and SmcHD1-mutant NSCs (Fig. 5a). We categorized the DNA-FISH signal localization relative to the *Xist* RNA cloud into four groups (Fig. 5b; see Methods). In wild-type NSCs, the majority of the SD/SI/CL signals were positioned similarly close to the Xi surface (Fig. 5b–c,e; inner/outer edge or protruded) or from the *Xist* cloud centroid (Fig. 5d,f), confirming earlier studies^{43,44}. However, the SD probes frequently protruded out of the *Xist* cloud (Fig. 5c,e) and became more distant from it (Fig. 5d,f) in SmcHD1-mutant NSCs. The SI (but not CL) probes exhibited the same trend but to a lesser extent (Fig. 5c–f).

To validate the protrusion and interaction of SD domains, we performed pair-wise DNA-FISH analysis using three SD probes (Fig. 5g). We observed that simultaneous protrusion of two SD probes was much more frequent in SmcHD1-mutant NSCs than in wild-type NSCs (Fig. 5h–i). Focusing on the ‘two-SD protrusion’ cells, the distance between 71-SD and 98-SD was significantly closer in SmcHD1-mutant NSCs (Fig. 5j–k), consistent with 4C-seq (Fig. 5g, pair-1). The distance between 98-SD and 148-SD was similar in SmcHD1-mutant and wild-type NSCs (Fig. 5j–k), again consistent with 4C-seq (Fig. 5g, pair-2 shows interaction in both WT and KO cells, although slightly higher in KO cells). The 71-SD and 148-SD signals didn’t become closer in SmcHD1-mutant NSCs (Fig. 5j–k), consistent with their weak interaction observed by 4C-seq (Fig. 5g, pair-3). Thus, while there is some cell-to-cell heterogeneity within the population, our FISH results were overall consistent with 4C-seq results (see Supplementary Text 1 for further discussion).

While our FISH data validated the 4C data, we admit that there is some degree of cell-to-cell heterogeneity. Please refer to the minor point 10 below on page 6 of this response letter (and Supplementary Text 1) for further explanation.

2. The authors suggest that the susceptibility of the SD domains to SmcHD1 KO is due to the spatial position of these domains at the edge of the chromosome territory and not due to preferential binding of SmcHD1 to these regions. To further support this hypothesis it would be helpful to see the SmcHD1 binding pattern in SD compared to SI domains by e.g. using (published) ChIP-seq or DamID data.

To test whether the defects present in the SD domains were due to the preferential binding of SmcHD1 to these domains, we analyzed the enrichment of SmcHD1 on the X chromosome using (1) DamID data of NPCs from Wang et al., Cell 2018 and (2) ChIP-seq data of MEFs from Gdula et al., Nat Comm 2019. We found that the binding pattern of SmcHD1 to SD domains differed between NPCs and MEFs, with SmcHD1 being enriched on SD domains in MEFs but not NPCs (Supplementary Fig. 8e). This suggests that SmcHD1 binding preference is cell-type-specific. However, the defects observed on the SD domains were relatively similar in

both cell types, for instance, as evidenced by the loss of H3K27me3 (Fig. 4a,d and Supplementary Fig. 8c–d). Thus, we concluded that the preferential binding of SmcHD1 to SD domains is probably not a major cause of the susceptibility of the SD domains to SmcHD1 KO. We described the results on page 10.

3. A previous study has identified regions on the Xi that are depleted for H3K27me3 in SmcHD1 KO cells (<https://doi.org/10.1038/s41467-018-07907-2>). Are these regions identical with SD domains?

Overall, we found that the SD domains also corresponded well with regions depleted of H3K27me3 in SmcHD1-mutant MEFs (Supplementary Fig. 8c–d). In addition, we observed that several SI domains, particularly those close to the telomere, also overlapped with H3K27me3-depleted regions in SmcHD1-mutant MEFs (Supplementary Fig. 8d). These results indicate that defects by SmcHD1 mutation are overlapping yet more extensive in MEFs (Gdula et al., 2019) than in NPCs. We described the results on page 10.

4. In Figure 3b, 4C-seq data from the Xi in NSCs on Day 9 shows increased contacts between the SD regions and weaker cis contacts compared to the other non-SD viewpoints. As the profiles are difficult to compare by eye, they should be quantified and analyzed statistically.

To show the increased contacts between the SD regions, we added the domainogram plots under the 4C-seq plots in Fig. 3b and Supplementary Figs. 5 and 6 to show the significant *cis* interactions using *p*-values. In addition, in order to demonstrate the relatively weaker *cis* contacts of the SD viewpoints in comparison to the non-SD viewpoints, we plotted the distance-dependent 4C-seq interaction scores between the viewpoints and the rest of the chromosome. Our analysis confirmed that there is a lower frequency of interactions with far-*cis* genomic regions, consistent with the idea that the SD viewpoints protruded out of the Xi chromosome territory. To make this point visually more convincing, we grouped the results together in Supplementary Figure 7 to show the overall trend.

5. The last section of the results is difficult to follow. It could improve the logical flow of the paper to integrate the Hi-C data (Fig. 6e–j) with the 4C-seq data in Figure 3, since both show SD-SD domain interactions. Here, a quantitative analysis of the SD domains identified by 4C-seq vs Hi-C could be useful.

We respectfully disagree with the idea of integrating the Hi-C data into Figure 3. In fact, it has bothered us for several years why the SD-SD domain interactions were not identified in the earlier Hi-C reports of SmcHD1-mutant cells in 2018. Just before the initial submission, we finally found that SD-SD interactions can be observed in the Hi-C data of Wang et al., 2018. Although this was in the form of a virtual 4C z-score heatmap we generated based on Wang et al. Hi-C data, it was reassuring to be able to finally see this, which prompted us to submit our article. Therefore, using this as an example, we would like to emphasize and caution that routine 4C/Hi-C analyses can overlook important features of chromosome architecture.

However, we agree with the reviewer that the section was complicated and difficult to follow. We reasoned that the rather unusual data representation in the form of a virtual 4C z-score heatmap could be one of the reasons why it was difficult for the reviewer to follow the logic. Therefore, we revisited the original Hi-C data of Wang et al., 2018, and after a careful re-examination, we finally managed to identify the SD-SD interactions in the Hi-C heatmap! Therefore, we moved the former Figure 6e–j to Supplementary Figure 10a–f and instead put more emphasis on the Hi-C heatmaps of wild-type and SmcHD1 mutant NPCs showing mutant-specific

emergence of SD-SD interactions in the new Figure 7e–f and described this in the main text.

In addition, we also wanted to emphasize that although the Xi is overall more compact compared to the Xa, our analyses suggest that they still share certain structural features, meaning that S1/S2 compartments are not necessarily unique to SmcHD1-mutant NPC Xi but such a feature is weakly present also on the wild-type NPC Xi in the form of PC3, not PC1, upon principal component analysis of Hi-C data.

We described all of this on pages 14–15 as follows. We hope these changes are satisfactory:

SD-SD domain interactions in SmcHD1-mutant cells are captured by Hi-C

One confounding enigma was that the previous Hi-C reports did not identify the SD-SD interactions in SmcHD1-mutant NPCs¹¹. However, as shown in Supplementary Fig. 8h, the virtual 4C data derived from these Hi-C data resembled our actual 4C results and showed SD-SD interactions, suggesting that Hi-C does capture what 4C captures. This led us to perform virtual 4C for viewpoints throughout the Xs in NPCs and plot their z-scores in a heatmap format (Supplementary Fig. 10a–f). From the z-score plots, we could immediately see strong local signals corresponding to SD-SD interactions on the Xi in the mutant but not wild-type NPCs (Supplementary Fig. 10b,d,f; circled regions). We thus revisited the original Hi-C heatmaps¹¹. By careful inspection, we could identify the SD-SD interactions on the Hi-C heatmap of the Xi in mutant but not wild-type NPCs (Fig. 7e; circled regions). Aggregated plots also revealed strong SD-SD interactions specifically in SmcHD1-mutant NPCs (Fig. 7f).

In addition, we found that the red/blue patterns of the z-score plots of the Xa resembled the Hi-C principal component 1 (PC1) profile to some extent (Supplementary Fig. 10a,c). This resemblance was also observed on the mutant NPC Xi (Supplementary Fig. 10d), perhaps reflecting the S1/S2 compartments^{11,51}. However, the red/blue patterns on the wild-type NPC Xi also resembled the mutant Xi, although the contrast was less enhanced (Supplementary Fig. 10b), suggesting the presence of S1/S2 compartments on the wild-type NPC Xi as well. Furthermore, when we calculated PC1 through PC4 on the Xa and Xi of various differentiation states and focused on those with >8% contribution rates, the PC3 of wild-type NPC Xi looked very similar to the S1/S2 compartments (Supplementary Fig. 10g–h). The same was true for the Xi in mouse Patski cells⁵² (Supplementary Fig. 10i).

Taken together, although the Xi is overall more compact compared to the Xa, our analyses suggest that they still share certain structural features. While the megadomain structure becomes prominent as the compact Xi is established (Supplementary Fig. 10g,j), the S1/S2 or A/B compartment features still remain in NPCs (Supplementary Fig. 10g–h). When the Xi heterochromatin is disturbed by perturbations such as SmcHD1 mutation, the relative contribution of different structural aspects of the Xi changes slightly, leading to, for instance, the emergence of S1/S2 compartments as PC1, not PC3, of the Hi-C contact map¹¹ or enhanced TAD boundaries in SmcHD1-mutant cells^{11,13}. In addition, our data demonstrate that new structural features, namely protrusion of SD domains and their interaction, emerge upon the loss of SmcHD1.

6. As mentioned in the introduction, the DXZ4 repeat region plays a structural key function on the inactive X chromosome. Therefore, it would be helpful to add the location of DXZ4 in the figures to assess its RT class and interaction pattern.

We added the location of *Dxz4* to the relevant figures.

7. *SmcHD1 is a well-studied protein which has been shown to play a role in X inactivation and higher-order chromatin architecture. It would be helpful to introduce its known functions from previous research already in the introduction, and also explain S1/S2 compartments.*

We made the following changes in the Introduction section on page 3 (in red):

With the advent of genome-wide chromosome conformation capture technology, Hi-C⁶, it is generally accepted that mammalian autosomes are composed of Mb-sized topologically-associating domains, TADs^{7,8}. TADs can be in either active (A) or inactive (B) nuclear compartments⁶, which are further subdivided into several subcompartments^{9,10}, although their significance remains largely unexplored. **When the Xi is formed during mouse embryonic stem cell (mESC) differentiation, it has been reported that neighboring small A/B compartment domains on the Xi initially fuse with each other to form larger S1/S2 compartment domains, which further merge to form the compact Xi structure through the actions of SmcHD1¹¹, a global Xi-binding protein known for its role in XCI maintenance^{11–15}. Unlike the autosomes, the Xi lacks TADs and compartments but instead has a unique bipartite ‘megadomain’ structure separated by a tandem macrosatellite repeat, *Dxz4/DXZ4*^{9,16–18}. While many molecular players involved in XCI have been identified, the details of how the Xi acquires this unique, compact 3D organization during XCI are still relatively unclear^{2,5}**

8. *In Fig. 2c, the number of domains in each category should be indicated.*

We added the number of domains in each category to the figure.

9. *We could not access the GEO repository.*

We confirmed that our GEO repository is accessible. Please access to our repository again using the following information.

To review GEO accession GSE211574:

Go to <https://www.ncbi.nlm.nih.gov/geo/query/acc.cgi?acc=GSE211574>

Enter token mfojqquvwpnyb into the box

Minor Comments

10. *In the model proposed in the study, SD domains protrude out of the Xi core, but at the same time cluster together. Does this happen in the same cell or in different cell subsets? It would be helpful to discuss in some more detail how this could be envisioned.*

As discussed earlier, while our FISH data validated the 4C data, there is cell-to-cell heterogeneity. Our data suggest that SD domain protrusion is frequent in SmcHD1-mutant NSCs and often simultaneous, allowing

multiple SD domains to interact with each other. However, the degree and/or frequency of protrusion may be different among SD domains, which may reflect the difference in the degree of RT reversal among different SD domains. That is, although single-cell Repli-seq revealed relatively uniform RT reversal of SD domains at the single-cell level (Fig. 2), the degree of RT reversal is slightly different among the SD domains. This difference may explain the differences in protrusion rate and interaction frequency among SD domains. As this is too complicated to discuss in the main text, we additionally discussed this point in Supplementary Text 1. We hope this is satisfactory.

11. In Fig. 5d, CE regions show the highest interchromosomal contacts (higher than SD domains). It would be useful to provide some more information on these regions and on how they differ from SD domains. What factors might keep them in the early replication phase?

We added the following phrase on page 12 of the main text (in red):

Virtual 4C supported the putative layered X chromosome organization in NPCs, with the SD and CE (CE contains *Xist*, which is highly expressed on the Xi; see also Supplementary Text 2) domains showing the highest inter-chromosomal interaction frequency, followed by SI, then CL domains on both Xs (Fig. 6c–d). The SD domains in mutant NPCs showed higher inter-chromosomal interaction frequency than in wild-type NPCs (Fig. 6c,e), consistent with the SD domains protruding out of the Xi core in the former. This chromosome-wide trend of SD domains preferentially making frequent inter-chromosomal interactions was conserved in mESCs (Fig. 6c–d).

In addition, we discussed this a bit more in the Supplementary Text 2, regarding the escapee genes within the CE domain. We hope these changes are satisfactory.

12. In Figure 1, it would help the reader's understanding to add a schematic of the Repli-seq method.

We added a schematic diagram of the Repli-seq method in Fig. 1b.

13. It would be useful to add the number of EtoL, LtoE, EtoE and LtoL domains and what percentage of the X chromosome they cover to Fig. 1.

We added the number of EtoL, LtoE, EtoE and LtoL domains and what percentages of the X chromosome they cover to Fig. 1c.

14. The term “% replication score” does not read well (line 136).

We made changes on pages 6–7 as follows (in red). We hope this is satisfactory:

To analyze the Xi's replication kinetics in detail, we performed scRepli-seq with cells throughout the S-phase that were ordered by their **percentage of replication (replication score)** to construct what we call the 'whole-S' RT profiles (Fig. 1e). In NSCs, the whole-S RT profile of the JF1-Xa looked similar to the

autosomes, as expected (Fig. 1e and Supplementary Fig. 2d). By contrast, the B6-Xi initiated replication in the second half of S-phase and lacked such clearly distinguishable early/late RT patterns (Fig. 1e).

While the **replication score progression** of the Xa was coordinated with that of the autosomes, it was not the case for the Xi (Fig. 1f–g and Supplementary Fig. 2e–f). The steep rise in the Xi's **replication score** of the fitted sigmoid curve (Fig. 1f, red line based on scRepli-seq average) suggested fast completion of the Xi replication with a $T_{10-90\%}$ (defined as the time required for a chromosome to go from 10% to 90% replication, assuming a 10-hr S-phase) of ~3.4 hr. By contrast, the $T_{10-90\%}$ of the Xa and the autosomes were significantly longer, on the order of ~8 hr (Fig. 1f and Supplementary Fig. 2e). The **replication scores** of the Xi were highly variable among cells within a given time window after 60–70% S phase (Fig. 1g). This suggests large variability in the timing of replication initiation among cells, and in turn, the timing of replication completion of the Xi. Thus, the Xi's $T_{10-90\%}$ of ~3.4 hr is probably an overestimate given that this was based on the scRepli-seq average.

15. The RT classes abbreviations should be explained in the legend of Figure 2, rather than Figure 3a.

The RT classes abbreviations in the legend of Figure 3a were removed and were instead added to the legend of Fig. 2a.

16. It should be indicated in the figure legend of Fig. 6e–j what the grey areas represent.

We moved the former Figure 6e–j to Supplementary Figure 10a–f and stated in the figure legend that gray areas represent low-coverage areas.

Decision Letter, first revision:

Message: Our ref: NSMB-A46855A

26th Apr 2023

Dear Dr. Hiratani,

Thank you for submitting your revised manuscript "Replication dynamics identifies the folding principles of the inactive X chromosome" (NSMB-A46855A). It has now been seen by the original referees and their comments are below. The reviewers find that the paper has improved in revision, and therefore we'll be happy in principle to publish it in Nature Structural & Molecular Biology, pending minor revisions to satisfy the referees' final requests (specifically, regarding statistical analysis of the FISH data, expanding on discussion of the implications of the FISH results, and to improve data representation and clarity) and to comply with our editorial and formatting guidelines.

We are now performing detailed checks on your paper and will send you a checklist detailing our editorial and formatting requirements in a couple of weeks. Please do not

upload the final materials and make any revisions until you receive this additional information from us.

To facilitate our work at this stage, it is important that we have a copy of the main text as a word file. If you could please send along a word version of this file as soon as possible, we would greatly appreciate it; please make sure to copy the NSMB account (cc'ed above).

Sincerely,
Sara

Sara Osman, Ph.D.
Associate Editor
Nature Structural & Molecular Biology

Reviewer #1 (Remarks to the Author):

The authors have convincingly answered the questions raised by the two referees. The article is therefore publishable in this new form.

Reviewer #2 (Remarks to the Author):

In the revised version of the paper the authors have added valuable data to confirm their main findings of the 3D organisation of the inactive X-chromosome through an orthogonal approach (DNA/RNA-FISH, Fig. 5) and evidence of SD-SD domain interaction in Hi-C maps (Fig. 7e). We support publication of the manuscript. Below you find some suggestions for minor changes to improve accuracy and clarity.

1. The authors of the study suggested a hierarchical organisation, where SmcHD1-dependent domains at the Xi's surface layer protrude out of the Xi core and interact upon SmcHD1 KO (Fig. 7g). In the revised version of the manuscript, the authors have validated part of this model with DNA/RNA-FISH at two loci (Fig. 5). At each locus, they used three different BAC probes targeting CL, SI and SD domains and located the DNA-FISH signal relative to the Xist cloud territory. With this approach they convincingly confirm the protrusion of SD domains with significantly increased SD probe localisation away from the Xist centroid in SmcHD1 KO NPCs (Fig. 5c-f). However, the data does not convincingly support the authors' hypothesis of SD domain interaction outside the Xi territory as they can only confirm increased proximity of protruded probes in one out of three SD probe pairs in KO cells compared to WT (Fig. 5j). Moreover, the distance of unprotruded SD probe pairs is similar or closer in both WT and KO cells compared to protruded probes (Fig. 5k). It is possible that SD domains show increased interaction within the Xist territory upon SmcHD1 KO, and this interaction must not necessarily occur outside of the core. Even though unprotruded probes do not show increased proximity in KO compared to WT NPCs in DNA-FISH experiments (Fig. 5j), the chromatin could be altered by SmcHD1 loss leading to punctual SD domain interaction which are visible in Hi-

C maps (Fig. 7e). We would suggest discussing the implications of the FISH data for the proposed model accordingly.

2. For the statistical analysis of the FISH data, multiple biological replicates were merged. It would be useful to also show the individual replicates for Fig. 5c and e in the Supplementary information to allow the reader to evaluate whether increased protrusion of SD domains in the mutant were indeed reproducible. Additionally, it would be helpful to add the p-values per category (e.g. "protruded") to show the significance of localisation changes of probes upon SmcHD1 KO.

3. To test whether the SD domains identified in NSCs in this study (which loose H3K27me3 in the SmcHD1-KO) overlap with previously identified domains in MEFs, which also loose H3K27me3 in the same KO, the authors have re-analyzed data from a previous study in MEFs (Supplementary Fig. 8c). They conclude that both cell types exhibit similar SD domain-specific loss of the repressive mark (Fig. 4a,d and Supplementary Fig. 8c-d). While this could be concluded from Fig. 4a and 4d for NPCs, it is not evident from Supplementary Fig. 8c where only a subset of SD domains seems to exhibit H3K27me3 loss in MEFs. We suggest adding the respective p-values and using violin plots instead of boxplots. Furthermore, the co-localisation of SD domains and regions with loss of H3K27me3 seems to be rather incomplete (Supplementary Fig. 8d). We would suggest discussing this point more clearly.

4. On p. 15 it is discussed that different levels of 3D organization can be seen in the different principal components (Fig. 10g-j). To allow the reader to follow this discussion, it would be useful to indicate SD domains, S1/S2 domains as well as A/B compartments and TAD boundaries in the plots in Fig. 10.

5. To allow other researchers to build on the findings, it would be useful to add a supplementary table with the genomic coordinates of SI, SD, CL and CI domains.

6. The addition of the Repli-seq method scheme in Fig. 1b is helpful. It would be useful to add a small text in the figure description and label the x-axis of the diagram.

7. In Fig. 7d, it would help the reader to label the y-axis in the same manner as 4b " $X_i/(X_i+X_a)$ ".

Author Rebuttal, first revision:

Point-by-point response to all of the reviewers' criticisms (reviewer comments in bold italic):

Reviewer #1:

Remarks to the Author:

The authors have convincingly answered the questions raised by the two referees. The article is therefore publishable in this new form.

We thank the reviewer for the positive comments and for understanding the value of our work.

Reviewer #2:

Remarks to the Author:

In the revised version of the paper the authors have added valuable data to confirm their main findings of the 3D organisation of the inactive X-chromosome through an orthogonal approach (DNA/RNA-FISH,

Fig. 5) and evidence of SD-SD domain interaction in Hi-C maps (Fig. 7e). We support publication of the manuscript. Below you find some suggestions for minor changes to improve accuracy and clarity.

1. The authors of the study suggested a hierarchical organisation, where SmcHD1-dependent domains at the Xi's surface layer protrude out of the Xi core and interact upon SmcHD1 KO (Fig. 7g). In the revised version of the manuscript, the authors have validated part of this model with DNA/RNA-FISH at two loci (Fig. 5). At each locus, they used three different BAC probes targeting CL, SI and SD domains and located the DNA-FISH signal relative to the Xist cloud territory. With this approach they convincingly confirm the protrusion of SD domains with significantly increased SD probe localisation away from the Xist centroid in SmcHD1 KO NPCs (Fig. 5c-f).

However, the data does not convincingly support the authors' hypothesis of SD domain interaction outside the Xi territory as they can only confirm increased proximity of protruded probes in one out of three SD probe pairs in KO cells compared to WT (Fig. 5j). Moreover, the distance of unprotruded SD probe pairs is similar or closer in both WT and KO cells compared to protruded probes (Fig. 5k). It is possible that SD domains show increased interaction within the Xist territory upon SmcHD1 KO, and this interaction must not necessarily occur outside of the core. Even though unprotruded probes do not show increased proximity in KO compared to WT NPCs in DNA-FISH experiments (Fig. 5j), the chromatin could be altered by SmcHD1 loss leading to punctual SD domain interaction which are visible in Hi-C maps (Fig. 7e). We would suggest discussing the implications of the FISH data for the proposed model accordingly.

We appreciated the reviewer's comments and discussed this alternative possibility in Supplementary Text 2 (referenced on page 9 of the main text). We also modified the description of our proposed model in Figure 7g legend (page 22) to reflect these points as follows:

Without SmcHD1, the Xi fails to maintain its late replication and 3D structure later during differentiation, resulting in frequent protrusion of SD domains located close to the surface of the Xi (but not SI domains), which occasionally interact with each other. While the figure shows the contact between protruded SD domains, it is also possible that SD-SD domain interactions could occur inside the Xi core (see Supplementary Text 2).

2. For the statistical analysis of the FISH data, multiple biological replicates were merged. It would be useful to also show the individual replicates for Fig. 5c and e in the Supplementary information to allow the reader to evaluate whether increased protrusion of SD domains in the mutant were indeed reproducible. Additionally, it would be helpful to add the p-values per category (e.g. "protruded") to show the significance of localisation changes of probes upon SmcHD1 KO.

We added the individual replicates for Fig. 5c and e in Supplementary Fig. 8f. We confirmed that results from individual replicates were consistent with the merged data. We also added p-values of the protruded group to each data, as shown in Fig. 5c,e and Supplementary Fig. 8f.

3. To test whether the SD domains identified in NSCs in this study (which loose H3K27me3 in the SmcHD1-KO) overlap with previously identified domains in MEFs, which also loose H3K27me3 in the same KO, the authors have re-analyzed data from a previous study in MEFs (Supplementary Fig. 8c). They conclude that both cell types exhibit similar SD domain-specific loss of the repressive mark (Fig. 4a,d and Supplementary Fig. 8c-d). While this could be concluded from Fig. 4a and 4d for NPCs, it is not evident from

Supplementary Fig. 8c where only a subset of SD domains seems to exhibit H3K27me3 loss in MEFs. We suggest adding the respective *p*-values and using violin plots instead of boxplots. Furthermore, the co-localisation of SD domains and regions with loss of H3K27me3 seems to be rather incomplete (Supplementary Fig. 8d). We would suggest discussing this point more clearly.

We replaced the original boxplots for H3K27me3 in MEFs with violin plots, as shown in Supplementary Figure 8c. To demonstrate that H3K27me3 enrichment on SD domains in SmcHD1-mutant cells was significantly lower than in wild-type MEFs, we added *p*-values obtained from the Wilcoxon signed-rank test with a one-tailed test.

Also, we discussed the incomplete loss of H3K27me3 as follows (page 8):

“The SD domains also coincided with regions depleted of H3K27me3 in SmcHD1-mutant MEFs¹³ (Supplementary Fig. 8c–d), **although parts of the SD domains retained H3K27me3. Several SI domains close to the telomere also showed H3K27me3 depletion** (Supplementary Fig. 8d).”

4. On p. 15 it is discussed that different levels of 3D organization can be seen in the different principal components (Fig. 10g–j). To allow the reader to follow this discussion, it would be useful to indicate SD domains, S1/S2 domains as well as A/B compartments and TAD boundaries in the plots in Fig. 10.

We added TAD boundary positions of mESCs (Dixon et al., 2012) and RT classes to Supplementary Fig. 10g,h,j. In addition, we added tracks of binarized A/B compartments of mESCs and binarized S1/S2 compartments of SmcHD1-mutant NPCs (both derived from Wang et al., 2018) to Supplementary Fig. 10h to help visualization of the data.

5. To allow other researchers to build on the findings, it would be useful to add a supplementary table with the genomic coordinates of SI, SD, CL and CI domains.

We provided the genomic coordinate information of SI, SD, CL, and CE domains in Supplementary Table 1. We also uploaded this information to our GEO: GSE211574.

6. The addition of the Repli-seq method scheme in Fig. 1b is helpful. It would be useful to add a small text in the figure description and label the x-axis of the diagram.

We labeled the DNA content on the x-axis of Fig. 1b. Also, we added a brief description of the figure in the legend (page 18) as follows:

(b) Schematic diagram of BrdU-IP Repli-seq experiment. BrdU-labeled cells are sorted into early and late S-phase fractions (gating strategy is shown in Supplementary Fig. 1i), which are subject to immunoprecipitation (IP) using an anti-BrdU antibody. The ratio of early and late S-phase DNA is calculated to generate a genome-wide RT profile.

7. In Fig. 7d, it would help the reader to label the y-axis in the same manner as 4b “ $X_i/(X_i+X_a)$ ”.

We modified the label on y-axis in Fig. 7d to become “Xi probability by RNA-seq [$X_i/(X_i+X_a)$].” We also corrected the label on y-axis in Fig. 4b in the same manner, as the previous version [X_i/X_i+X_a] was not an appropriate formula.

Final Decision Letter:**Message** 28th Jun 2023

:

Dear Dr. Hiratani,

We are now happy to accept your revised paper "Replication dynamics identifies the folding principles of the inactive X chromosome" for publication as a Article in Nature Structural & Molecular Biology.

As soon as your article is published, you can generate your shareable link by entering the DOI of your article here: http://authors.springernature.com/share. Corresponding authors will also receive an automated email with the shareable link

Your paper will be published online soon after we receive proof corrections and will appear in print in the next available issue. You can find out your date of online publication by contacting the production team shortly after sending your proof corrections. Content is published online weekly on Mondays and Thursdays, and the embargo is set at 16:00 London time (GMT)/11:00 am US Eastern time (EST) on the day of publication. Now is the time to inform your Public Relations or Press Office about your paper, as they might be interested in promoting its publication. This will allow them time to prepare an accurate and satisfactory press release. Include your manuscript tracking number (NSMB-A46855B) and our journal name, which they will need when they contact our press office.

About one week before your paper is published online, we shall be distributing a press release to news organizations worldwide, which may very well include details of your work. We are happy for your institution or funding agency to prepare its own press release, but it must mention the embargo date and Nature Structural & Molecular Biology. If you or your Press Office have any enquiries in the meantime, please contact press@nature.com.

Please note that *Nature Structural & Molecular Biology* is a Transformative Journal (TJ). Authors may publish their research with us through the traditional subscription access route or make their paper immediately open access through payment of an article-processing charge (APC). Authors will not be required to make a final decision about access to their article until it has been accepted. <https://www.springernature.com/gp/open-research/transformative-journals> Find out more about Transformative Journals

Authors may need to take specific actions to achieve

<https://www.springernature.com/gp/open-research/funding/policy-compliance-faqs> **compliance** **with funder and institutional open access mandates**. If your research is supported by a funder that requires immediate open access (e.g. according to [Plan S principles](https://www.springernature.com/gp/open-research/plan-s-compliance)) then you should select the gold OA route, and we will direct you to the compliant route where possible. For authors selecting the subscription publication route, the journal's standard licensing terms will need to be accepted, including [self-archiving policies](https://www.springernature.com/gp/open-research/policies/journal-policies). Those licensing terms will supersede any other terms that the author or any third party may assert apply to any version of the manuscript.

Sincerely,
Sara

Sara Osman, Ph.D.
Associate Editor
Nature Structural & Molecular Biology